# Personalized customization: Service resource configuration optimization driven by customer requirements accurately

**Chao Yu, Haibin Wang** *

School of Management, Shenyang University of Technology, Shenyang, China

* wanghaibin82596@163.com

## Abstract

Proposing an approach of service resource configuration optimization driven by customer requirements to address the issue of service resource configuration optimization in the context of personalized customization. Firstly, the importance judgment matrix, KANO model, and competitiveness evaluation are integrated to evaluate the relative importance of customer requirements. Secondly, the House of Quality (HoQ) and the intermediary variable "technical attributes" are utilized to determine the weight of each service module and its correlation with customer requirements. Afterwards, due to the varying customer requirements, the service candidate itemsets under the same service module will differ. To address this, a "one-to-many" relationship mechanism is introduced between the service module and service candidate itemsets. The service candidate itemsets are determined based on the correlated customer requirements. On this basis, the customer's perceived utility is determined by applying the four types of utility measure functions. The service resource configuration scheme is established by formulating and solving an optimization model. Finally, the viability and efficacy of the approach are demonstrated with an example of living room customization by a customization company, utilizing an improved genetic algorithm (IGA).

## 1. Introduction

In the current era of personalized customization, the market has shifted towards a "buyer's market," where customer diversity and personalized requirements are increasingly prominent. This poses significant challenges for traditional manufacturing companies, as they must balance the contradiction to meet individual customer requirements while also delivering products quickly and cost-effectively. To address this, enterprises must shift away from the traditional production mindset of "factory to customer" and instead adopt a "user-driven" approach to personalized production. This shift from product-oriented to service-oriented enterprises has become an inevitable trend in enterprise development [1]

The C2B2M business model based on mass customization is a business model that carries out mass customized production according to customer requirements, from customers to enterprises to factories. It is a business model supported by modern information technology and based on massive customer requirements through modular design and parts

**Data availability statement:** All relevant data are within the manuscript and its Supporting Information files.

**Funding:** Funding statement: This work was supported in part by the Liaoning Provincial Social Science Planning Fund [L21CGL021].

**Competing interests:** The authors have declared that no competing interests exist.

standardization to improve customer satisfaction with personalized requirements and improve the service efficiency of enterprise production [2]. The essence of mass customization is to identify the individualized requirements of customers and offer a tailored service resource configuration scheme. One of its core contents is service resource configuration optimization [3].

To begin with, service requirements analysis, as the source of service resource configuration optimization, mainly includes acquiring, categorizing, and evaluating service requirements. Based on this analysis, the relationship between service requirements and technical attributes or service modules is established [4]. Among them, service requirements importance evaluation has always been an important part of studying service requirements. Accurately evaluating the importance of customer requirements and establishing their priority accordingly can enhance the correlation of the service resource configuration scheme with the actual customer requirements. Hence, the matter of precisely acquiring the requirements of each customer and evaluating their importance remains deserving of consideration.

Furthermore, what requires attention is that various service resource configuration schemes will exhibit varying levels of performance in satisfying customer requirements. This directly impacts customer satisfaction with the service resource configuration schemes and subsequently influences the company's revenue [5].

Accurate correlation and matching between service resources and customer requirements are essential to the service resource configuration scheme. Typically, customers will have specific expectations and requirements for service resource configuration scheme based on their individual circumstances. These expectations and requirements will serve as criteria to evaluate their satisfaction with the service resource configuration scheme and the level of satisfaction they experience. Simply speaking, customers' perceived utility of configuration schemes will vary depending on the scheme [6]. Recent research has demonstrated that employing modular design for service resource configuration schemes is an efficient approach to addressing this issue [7].

The service resource configuration scheme matches customer requirements through the correlation of service modules and customer requirements. This ensures that customer expectations are met to the fullest extent and maximizes the customer's perceived utility under certain constraints. Hence, in the process of service design, it is imperative to take into account the match of each service module with customer requirements in the service resource configuration scheme. However, in conventional research, there is typically a one-to-one relationship mechanism between service modules and their candidate itemsets. In this case, if a service module is correlated with multiple requirements, a situation may occur where the candidate itemset exhibits a high level of matching when connected with one requirement but lacks sufficient matching when correlated with another requirement. Consequently, the service resource configuration scheme has challenges in accurately and objectively fulfilling customer requirements and preferences.

Presently, introducing a "one-to-many" relationship mechanism between the service module and its candidate itemset will effectively improve or resolve this issue. Fig 1 provides a detailed example illustrating the concept of service modules and their candidate itemsets within the service resource configuration framework, as well as their "one-to-many" relationship mechanism. The Air Conditioner, functioning as a service module, will choose suitable candidate items from its various candidate itemsets to fulfill different customer requirements. Ultimately, it will generate a candidate item configuration scheme for the Air Conditioner. Thus, how to extract customer requirements and evaluate the importance of customer requirements, modularize the service resource configuration scheme, divide the candidate itemsets under the module, establish the corresponding relationship between the

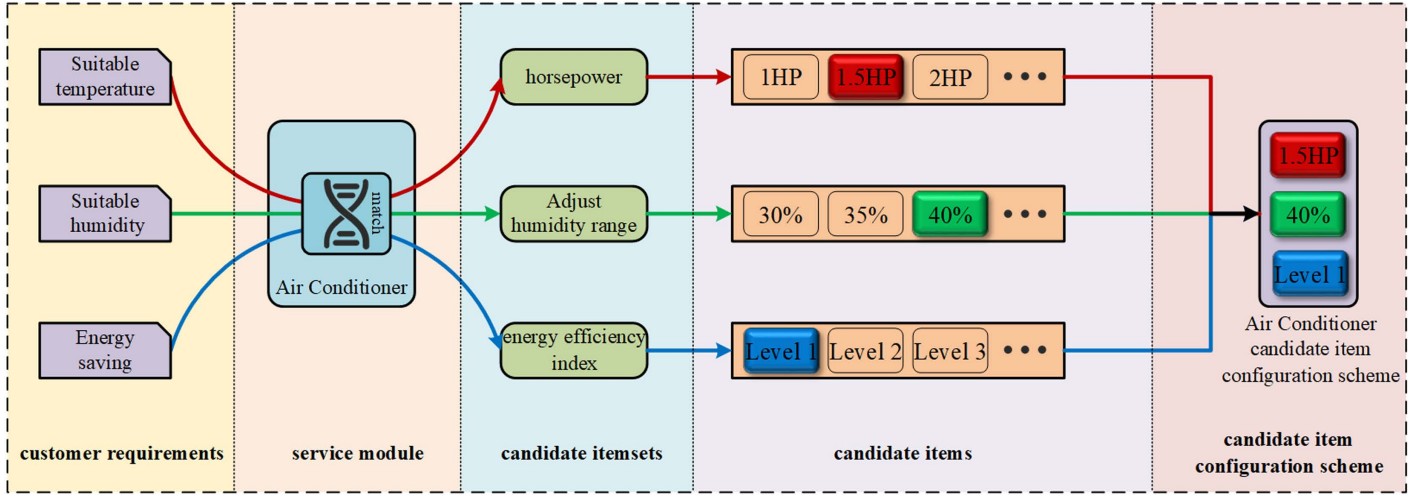

**Fig. 1. Service resource configuration structure: an example.**

service module and the candidate itemset, and then realize an effective correlation between the service module and the customer requirements, and on this basis, the issue of supporting or assisting service-oriented enterprises in effectively optimizing the service resource configuration scheme is also worthy of attention.

Considering the circumstances, the objective of this article is to present a service resource configuration optimization method driven by customer requirements. This method employs a combination of the importance judgment matrix, the KANO model, and competitiveness evaluation to quantitatively evaluate the importance of consumer requirements. On this basis, we apply the House of Quality of QFD and integrate it with the intermediary variable " technical attributes" to evaluate the importance of each service module and the relationship between the service module and customer requirements. Afterward, it is considered that the candidate itemsets within the same service module would vary when the service module is correlated with different customer requirements. Consequently, the different service candidate itemsets associated with the customer requirements under the service module are defined. Next, the customer's perceived utility is computed using the four types of utility measure functions. Additionally, we determine the service resource configuration scheme by formulating and solving the optimization model. Finally, taking the living room customization by a customization company as an example and using the improved genetic algorithm (IGA) to illustrate the feasibility and effectiveness of the method.

The subsequent parts of this article are arranged as follows: Section 2 briefly reviews the relevant literature on research regarding service resource configuration optimization methods. Section 3 describes the service resource configuration optimization issues studied in this article and defines parameters; Section 4 explains the basics of service resource configuration optimization based on customer requirements, and calculation steps; Section 5 is an introduction to the improved genetic algorithm(IGA); Section 6 is a case analysis and algorithm comparison; and finally, Section 7 is the conclusion of the full article.

## 2. Related work

Currently, the available research has made progress in the field of service resource configuration optimization. However, there is a scarcity of studies that consider the "one-to-many" relationship mechanism between service modules and their candidate itemsets. Prior study

findings often center around three key areas: analysis of service requirements, modularization and processing of service resources, and configuration of service resource schemes. The specific results of relevant research are as follows:

In terms of service requirement analysis, customer requirements have always been the source of corresponding resource configuration analysis in service resource configuration optimization. We must fully acquire, categorize, screen, and evaluate customer requirements to ensure that the provided service resource configuration scheme meets them more efficiently. In contrast to conventional methods of requirement acquirement, such as questionnaires and telephone interviews, Wang et al. [8] proposed a graph-based context-aware requirement elicitation approach considering contextual information within the Smart PSS to extract implicit stakeholder requirements within a specific context. Ali et al. [9] proposed a social network service-based requirement engineering process. It considers the attributes of users' opinions to determine variability and commonality. Chen et al. [10] proposed a hybrid framework integrating the rough-fuzzy best-worst method to identify and prioritize the customer activity-oriented service requirements for a smart product service system. Kayapinar et al. [11] proposed a service-quality approach connected with SERVQUAL and fuzzy QFD to determine customers' requirements and then presented the FMODM for prioritizing design characteristics to minimize the technical difficulty and maximize the total weights of technical design requirements. Bi et al. [12] proposed a method for modeling customer satisfaction from online reviews. In the method, customer satisfaction dimensions (CSDs) are first extracted from online reviews based on latent Dirichlet allocation (LDA). The sentiment orientations of the extracted CSDs are identified using a support vector machine (SVM). Then, proposed an ensemble neural network-based model (ENNM) to measure the effects of customer sentiments toward different CSDs on customer satisfaction. Bi et al. [13]proposed an ensemble deep learning method for forecasting daily tourism demand for tourist attractions with big data to fully capture the relationship between these forecasting variables and actual tourism demand automatically. Jin et al. [14] proposed a method for mining online reviews with a Kansei-integrated Kano model for innovative product design. Enlightened by the Kano model, product features are prioritized based on affective emotions to show their importance to customer satisfaction. Aydin et al. [15] proposed a sustainable linear programming (LP)-based Quality Function Deployment methodology under the IVIF environment. The proposed method determines more accurate customer expectations and related service requirements. Sun et al. [16] proposed a Customer-Manufacturer-Kano (CM-Kano) model to analyze the ever-changing opinions of customers and manufacturers to provide improvement strategies for product design. Zhou et al. [17] proposed a framework of user experience-oriented smart service requirement (UXO-SSR) analysis for smart product service system development, which can provide an effective tool for acquiring the UX requirements of smart PSS from a holistic perspective.

In terms of service resources modularization, it is an effective approach to addressing the conflict between the diverse customer requirements and the service providers' requirements for standardization and efficiency enhancement. It is widely used in industries like tourism, medical care, and finance. Zhang et al. [18] proposed a method to partition the service flow module based on fuzzy spaces quotient theory to reduce the subjectivity of module granularity selection in the research of modular design of service flow. Böttcher et al. [19] utilized propositional logic and linear temporal logic to examine the logical and temporal interdependencies among modules. Zhang et al. [20] proposed a Design Structure Matrix (DSM) as a technique for healthcare process modularization., and developed a DSM-based modularization and sequencing algorithm to support modular clinical pathway design. Ji et al. [21] proposed a green modular design for material efficiency that can facilitate life-cycle material efficiency

through component material reuse and minimize resource commitment throughout the product realization process. Moon et al. [22] proposed a module-based service model to facilitate customized service design and represent the relationships between functions and processes in a service. They considered a module selection problem for platform design as a strategic module-sharing problem in a collaboration situation. Tuunanen et al. [23] proposed a modular service design framework and a service design method that adopts DPs to create effective modular ITeS designs and also offered ways to conceptualize and apply service modularization to improve the adoption of modular service design by service designers and managers. Chung et al. [24] conceptualized modularity as multidimensional and investigated how these multidimensional SDK-based modularity choices impact the performance of a key category of digital products—mobile apps. Van et al. [25] proposed that solution modularity is a critical mechanism for enhancing cross-selling opportunities. Ultimately, this can help the company achieve the optimal level of cross-selling opportunities, thereby facilitating its growth. Sheng et al. [26] identified four equifinal configurations sufficient for high MCC and categorized them into three types: modularity + integration oriented, integration + customer need oriented, and modularity + integration + customer need balance. Kang et al. [27] developed a moderated mediation model in which quality-oriented product design practices influence operational performance via supplier involvement under the different levels of product modularity.

In terms of service resource scheme configuration, based on the modularization of service resources, the service module content can be configured to generate a service resource configuration scheme that optimally fulfills customer requirements. Luo et al. [28] established a service configuration optimization model for multiple customers, considering service process information and time-varying service resource constraints based on quality function deployment and modularization. Li et al. [29] developed an SDF-oriented genetic algorithm to effectively create a manufacturing service composition with large-scale candidate services. Zhang et al. [30] studied product-oriented product–service system configuration optimization from a fine-grained perspective. A multilayer network composed of (i) a product layer, (ii) a service layer, and (iii) a resource layer was constructed to represent the elements (product parts, service activities, and resources) and relationships in PSS. Service activity selection and resource allocation were considered jointly to construct the mathematical model of product-service system configuration optimization. Wang et al. [31] proposed a method that adopts the digital twin and augmented Lagrangian coordination (ALC) to perform service model construction and optimal configuration of shared manufacturing resources. For the resource configuration and scheduling algorithm, Zhang et al. [32] proposed a dynamic priority based on the dominant resource proportion and valid active time to improve social welfare and resource utilization. Al-Wesabi et al. [33] presented new hybrid metaheuristics for energy efficiency resource allocation (HMEERA) for the cloud computing environment to achieve the optimal configuration of resources. Yang et al. [34] created a four-step generic product service system design method that is based on service requirements. The steps are smart and connected service product configuration, dynamic event-state knowledge graph-based service activity flow, and service resource network configuration. Nie et al. [35] established the cloud production line model to support the optimal configuration of the distributed idle manufacturing resources by applying a systematic evaluation method and digital twin technology, which reflect the actual manufacturing scenario of the whole production process. Dong et al. [36] proposed a novel Smart Product-service System configuration method oriented to Mass personalization to meet the personalized and dynamic requirements of customers, especially satisfy affective and functional requirements simultaneously throughout the reconfiguration life-cycle. Gai et al. [37] focused on the resource configuration issue in the Internet of Things and proposed a novel approach that uses a Reinforcement Learning mechanism to construct

the strategy of resource configuration. Implementing a Reinforcement Learning mechanism intends to prevent contradictions and make resource configuration operations smart.

In conclusion, the available researches on service resource configuration optimization have achieved certain results in the three key areas of service requirements analysis, service resources modularization and processing, and service resource schemes configuration. Nevertheless, after associating the service modules in the service resource configuration scheme with customer requirements, it is rarely seen that "the service candidate itemset under the same service module will be different due to different customer requirements met by the module. The candidate itemset within the same service module will be different due to the different customer requirements met by the module; this means that there is a "one-to-many" relationship mechanism between the service module and its candidate itemsets, rather than just a "one-to-one" relationship mechanism. Hence, when modularizing the service resource configuration scheme, it is more practical and better aligned with customer expectations to take into account the "one-to-many" relationship mechanism between the service module and its candidate itemsets, which refers to situations where different candidate itemsets may exist under the same service module.

## 3.  Problem description

This study, as depicted in Fig 2, concentrates on the problem of service resource configuration optimization driven by customer requirements. In this case, the personalized service platform engages in focused conversation and research with customers to dig out their requirements for service resource configuration scheme and, on this basis, uses appropriate mathematical methods to quantify and rank the importance of customer requirements; The platform divides service resource configuration scheme into several service modules. Each service module has several (at least one) service candidate itemsets. Each service candidate itemset contains several service candidate items.

In actual applications, the same customer requirement may require multiple service modules in the service resource configuration scheme to satisfy it. Similarly, the same service module may also satisfy multiple customer requirements. At this time, the candidate itemset within the same service module will be different due to the different customer requirements met by the module; this means that there is a "one-to-many" relationship mechanism between the service module and its candidate itemsets. For example, in the whole customization process, when choosing an Air Conditioner, if you are concerned about the customer's refrigeration and heating requirements, you need to pay attention to the air conditioner's horsepower selection. If you are concerned about the customer's energy-saving requirement, you need to consider the air conditioner's energy efficiency index selection. When satisfying different requirements, the air conditioning service module selects different candidate itemsets, namely the horsepower itemset and the energy efficiency index itemset. The satisfaction of customers' various requirements is influenced by different factors under Air Conditioner.

Hence, to satisfy customer requirements more accurately and efficiently, it is particularly important to select diverse candidate items, measure utility, and integrate them based on service modules for the same service module when satisfying different customer requirements. The problem to be solved in this study is how to accurately identify and evaluate customer requirements, and on this basis, select and integrate appropriate candidate items for different service modules under the conditions of satisfying different customer requirements and cost constraints, so as to determine the service resource configuration scheme.

The following are the sets and variables involved in the service resource configuration optimization problem driven by customer requirements that this study focuses on.

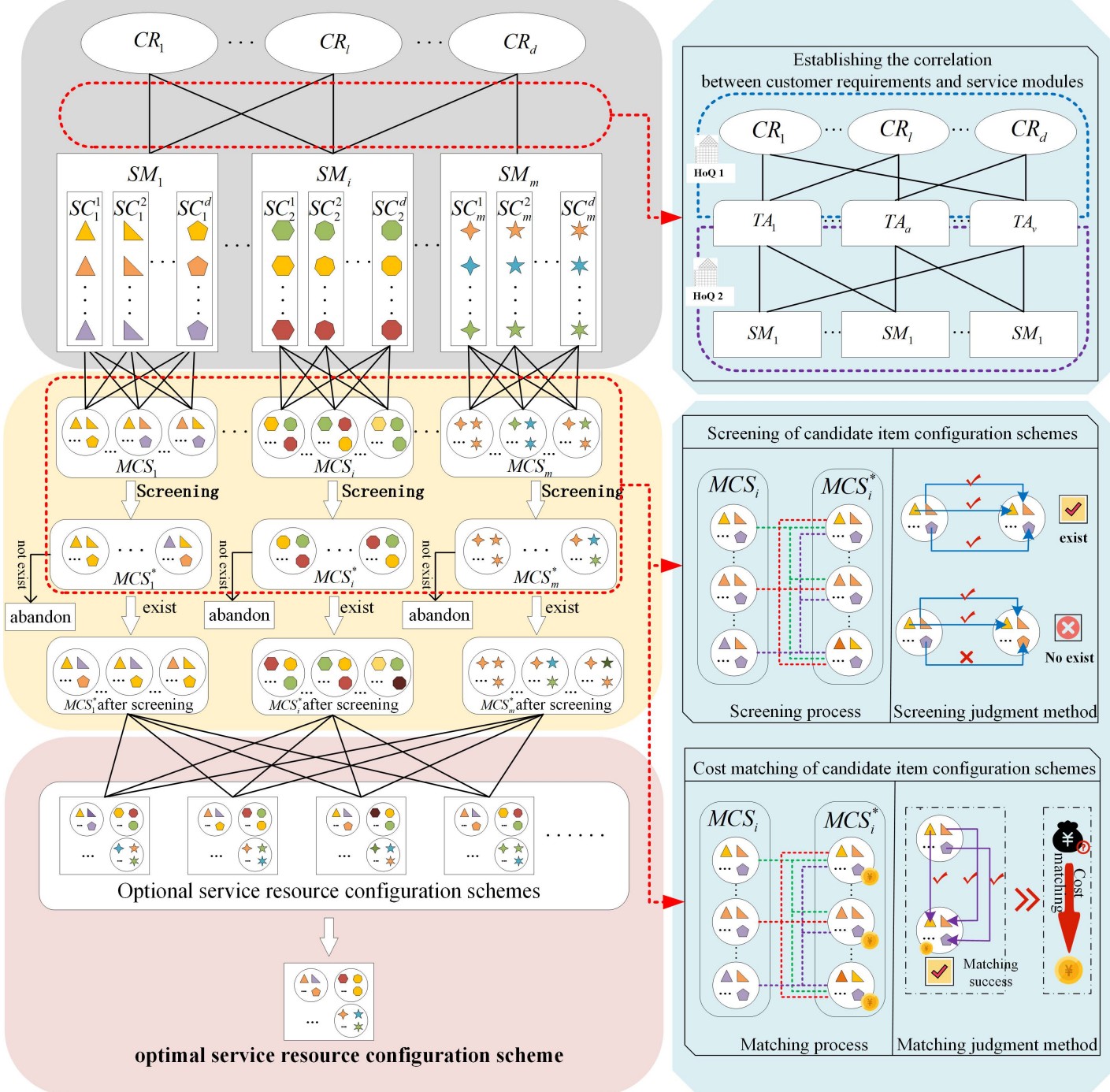

**Fig 2. Service resource configuration optimization problem driven by customer requirements accurately.**

$CR = \{CR_1, CR_2, ..., CR_d\}$ is the customer requirement set, where $CR_l$ represents the $l$-th customer requirement, $l = 1, 2, ..., d$.

$TA = \{TA_1, TA_2, ..., TA_v\}$ is the technical attribute set, where $TA_a$ represents the $a$-th technical attribute, $a = 1, 2, ..., v$.

$SM = \{SM_1, SM_2, ...., SM_m\}$ is the service module set, where $SM_i$ represents the $i$-th service module in the service resource configuration scheme.

$SC_i^l = \{SC_{i1}^l, SC_{i2}^l, ..., SC_{in_i^l}^l\}$ is the candidate itemset of service module $SM_i$ when satisfying the customer requirement $CR_l$, where $SC_{ij}^l$ represents the $j$-th candidate item of service module $SM_i$ when satisfying the customer requirement $CR_l$, $i = 1, 2, ..., m$, $j = 1, 2, ..., n_i^l$, $l = 1, 2, ..., d$.

$MCS_i = \{OC_{i1}, OC_{i2}, ..., OC_{ih^i}\}$ is all candidate item configuration schemes set in the service module $SM_i$, where $OC_{ik} = \{SC_{ij}^1, SC_{ij}^2, ..., SC_{ij}^d\}$ represents the k-th candidate item configuration scheme in the service module $SM_i$, and $SC_{ij}^l$ represents the $j$-th candidate item of service module $SM_i$ when satisfying the customer requirement $CR_l$. If service module $SM_i$ is correlated with the requirement $CR_l$, $SC_{ij}^l$ is a specific candidate item, otherwise, $SC_{ij}^l$ is $\varnothing$, $i = 1, 2, ..., m$, $k = 1, 2, ..., h^i$, $j = 1, 2, ..., n_i^l$, $l = 1, 2, ..., d$.

$MCS_i^* = \{OC_{i1}^*, OC_{i2}^*, ..., OC_{it^i}^*\}$ is feasible candidate item configuration schemes set in the service module $SM_i$, where $OC_{is}^*$ represents the $s$-th feasible candidate item configuration scheme in the service module $SM_i$, $i = 1, 2, ..., m$, $s = 1, 2, ..., t^i$, $t^i \leq h^i$.

$w_l$ is the relative importance weight of the customer requirement $CR_l$.

$c_i$ is the cost of the service module $SM_i$.

$c_{fix}$ is the fixed cost during the service process.

$y_{kij(si^*j^*)}^l$ is a "0-1" variable, which is used to represent the matching relationship between the candidate item configuration scheme and the feasible candidate item configuration scheme in the service module $SM_i$. If candidate items $SC_{ij}^l$ in the candidate item configuration scheme $OC_{ik}$ are all consistent with candidate items $SC_{i^*j^*}^l$ in $OC_{is}^*$, $y_{kij(si^*j^*)}^l = 1$, otherwise, $y_{kij(si^*j^*)}^l = 0$.

$x_{ij}^l$ is a "0-1" decision variable, if candidate items $SC_{ij}^l$ is selected, $x_{ij}^l = 1$, otherwise, $x_{ij}^l = 0$.

The problem to be addressed in this study is: based on the relevant decision-making information such as $CR$, $TA$, $SM$, and $SC$, combined with the research method of this study, solve the problem of service resource configuration optimization driven by customer requirements; $MCS$ and $y$ are sets or variables utilized to facilitate solution screening.

## 4. Theory and methods

What needs to be clear is this article only deals with the issue of service resource configuration optimization in the context of personalized customization, investigates the customer's demand for decoration industry and gives their importance judgments. There is no ethics involved. Survey of the requirements of customers covered in this example was conducted in May-June 2024. Participants were provided with the right to information and verbal consent was given by the participant, which was witnessed with a colleague who was investigating with them. This survey investigated 148 adult customers, not involving minors, and distributed questionnaires to them, mainly to investigate their requirements for the decoration industry and give their importance judgments. In this process, these 148 adult customers mainly participated in the filling of KANO model questionnaire for calculating the loss cost importance.

The following is a detailed description of the fundamental principles and calculation steps for service resource configuration optimization driven by customer requirements. Fig 3 illustrates the flow chart of specific process of service resource configuration optimization problem.

### 4.1. Quantification of the relative importance weight of customer requirements based on QFD

During the service process, in order to make the service resource configuration scheme better satisfy the personalized customer requirements, before formulating the corresponding

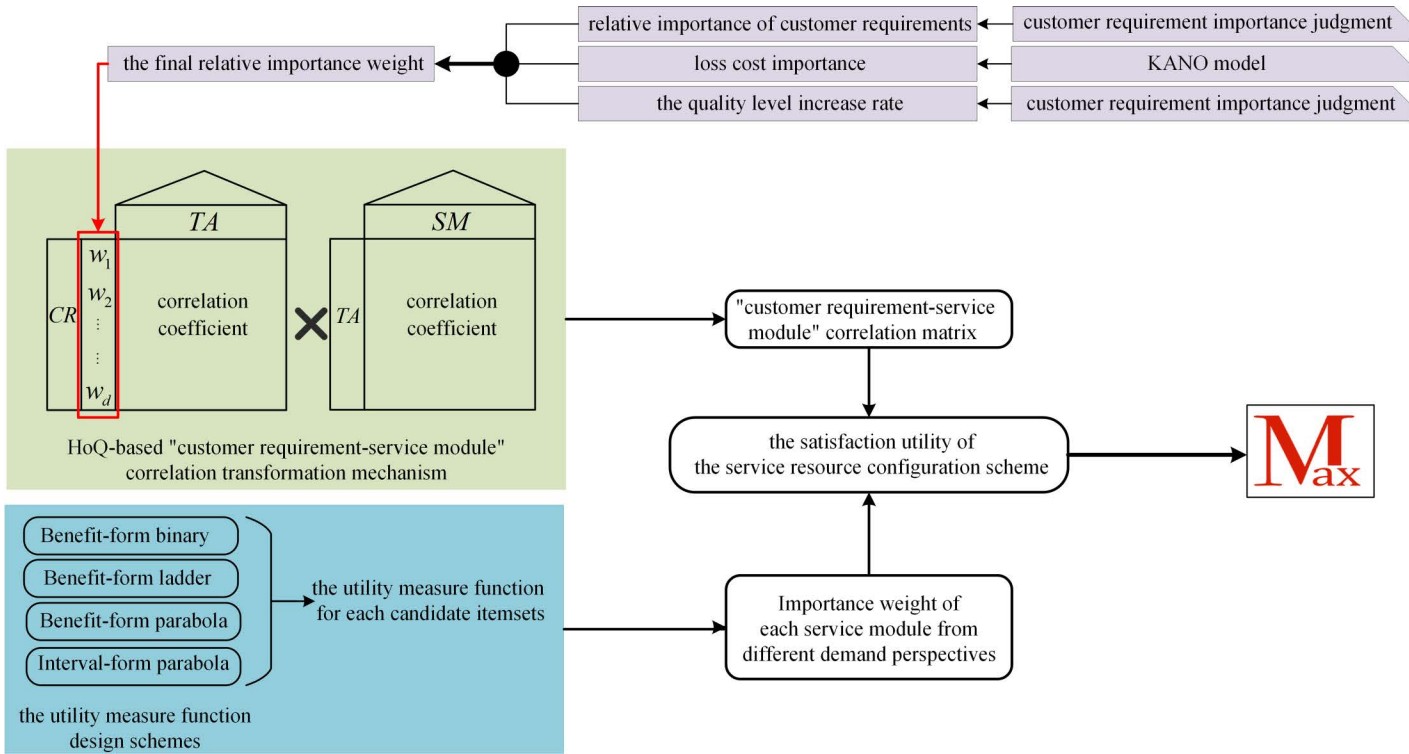

**Fig 3. Flow chart of specific process of service resource configuration optimization problem.**

service scheme, it is necessary to conduct targeted investigation and research on customer requirements, quantify the importance weight of each customer requirement, and determine its priority. In actual applications, a customer requirement may require multiple service modules to jointly satisfy; that is, there is a certain correlation between the two. We use Quality Function Deployment (QFD) to refine customer requirements into specific technical attributes, establish a correlation between customer requirements and service modules, and determine the relative importance weight of each service module. Fig 4 illustrates the transformation relationship between customer requirements, technical attributes, and service modules.

Quantification and prioritizing of the relative importance of customer requirements. To establish the correlation relationship between customer requirements and service modules, this study introduces the House of Quality (HoQ) in QFD into the research. During the process of constructing the HoQ, it is necessary to quantify and prioritize the importance of customer requirements. When the number of customer requirements is within a small range, the human brain's prioritization of their importance is reasonable. We can let customers directly prioritize the importance of each requirement through questionnaires. However, in a comprehensive service process, there will be a substantial number of requirements that need to be studied. When the number of requirements is large, it is difficult for customers to objectively and reasonably make accurate judgments and prioritize the importance of each requirement. Taking this into account, the study utilized a method that combines three approaches: customer requirement importance judgment matrix, KANO model, and competitiveness evaluation. We used this method to prioritize and evaluate the importance of each customer requirement, as follows:

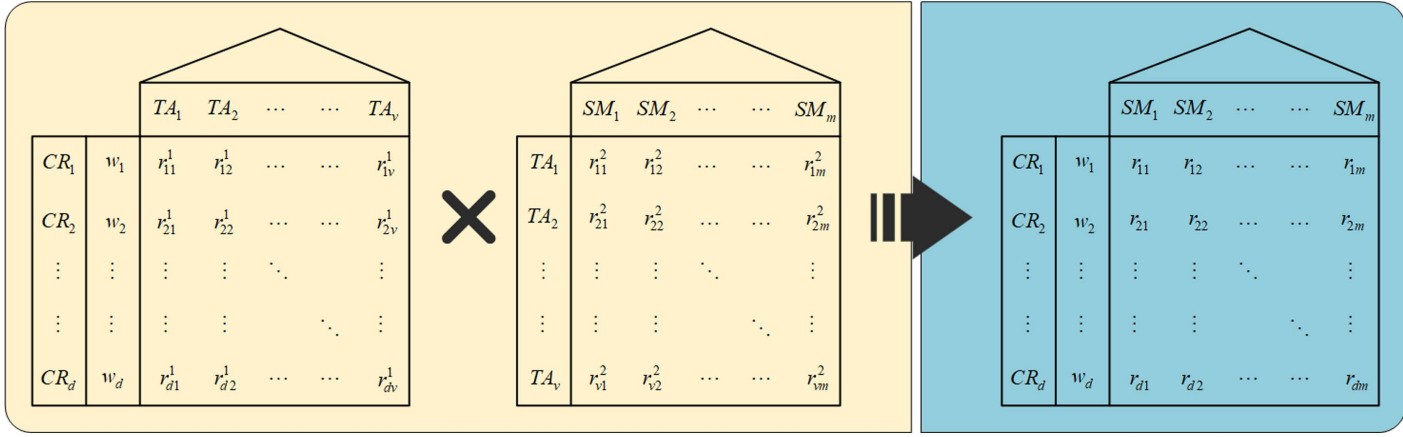

**Fig 4. HoQ-based "customer requirement-service module" correlation transformation mechanism.**

**Step 1:** Requirement importance determination based on the requirement importance judgment matrix. Customers have varying priorities for their requirements, so it is necessary to evaluate their importance. In this study, we initially collect, categorize, and organize the customer requirements using the hierarchical analysis method to make the collected requirements more detailed and differentiated. Subsequently, on this basis, we utilize the customer requirement importance judgment matrix to evaluate the importance of customer requirements, as follows:

(1) Using the customer requirement importance judgment matrix to compare the collected customer requirements and generate an importance judgment matrix, as illustrated in Table 1.

Thus, construct an importance judgment matrix $Z = \left[ z_{ll^*} \right]_{d \times d}$, where $z_{ll^*}$ represents the importance of customer requirements $CR_l$ compared to $CR_{l^*}$, $z_{ll^*} = \dfrac{1}{z_{l^*l}}$, $l = 1,2,...,d$, $l^* = 1,2,...,d$. Table 2 below displays the specific numerical criteria for $z_{ll^*}$.

(2) Calculation of customer requirement importance based on the customer requirement importance judgment matrix.

The formula for calculating the relative importance of customer requirements based on the customer requirements importance judgment matrix is as follows:

$$w_l^Z = \frac{\overline{w_l^Z}}{\sum\limits_{l=1}^{d} \overline{w_l^Z}} \tag{1}$$

where $\overline{w_l^Z}$ is the absolute importance weight of each requirement based on the customer requirement importance judgment matrix, and the calculation formula is as follows:

$$\overline{w_l^Z} = \sqrt[d]{\prod_{l^*=1}^{d} z_{ll^*}} \tag{2}$$

**Step 2:** Calculation of loss cost importance based on the KANO model. The KANO model categorizes customer requirements into attractive requirements, one-dimensional requirements, must-be requirements, indifferent requirements, and reverse requirements. We categorize customer requirements by making and distributing KANO model questionnaires, fully

**Table 1. The customer requirement importance judgment matrix.**

| CR | CR | | | | |
|---|---|---|---|---|---|
| | $CR_1$ | $CR_2$ | ... | ... | $CR_d$ |
| $CR_1$ | $z_{11}$ | $z_{12}$ | ... | ... | $z_{1d}$ |
| $CR_2$ | $z_{21}$ | $z_{22}$ | ... | ... | $z_{2d}$ |
| $\vdots$ | $\vdots$ | $\vdots$ | $\ddots$ | | $\vdots$ |
| $\vdots$ | $\vdots$ | $\vdots$ | | $\ddots$ | $\vdots$ |
| $CR_d$ | $z_{d1}$ | $z_{d2}$ | ... | ... | $z_{dd}$ |

**Table 2. The specific numerical criteria for. $z_{ll^*}$**

| Meaning (the former compared to the latter) | Equally important | Slightly important | Important | Very important | Absolutely important |
|---|---|---|---|---|---|
| Numerical criteria | 1/1 | 2/1 | 3/1 | 4/1 | 5/1 |

utilizing the KANO model's advantages in identifying customer requirements across various dimensions. According to the differences in customer requirements across different categories, the requirement elements that can cause customer dissatisfaction are focused on and evaluated. Table 3 illustrates the specific numerical criteria of customer requirement importance in the KANO model.

Based on the established requirement importance numerical criteria and the questionnaire results of the KANO model, the calculation formulas for absolute importance and loss cost importance can be obtained as follows:

$$AI_l = o_l^{MR} \times 5 + o_l^{OR} \times 3 + o_l^{AR} \times 1 + o_l^{IR} \times 0 - o_l^{RR} \times 3 \tag{3}$$

$$w_l^{LCI} = \frac{AI_l}{\sum_{l=1}^{d} AI_l} \tag{4}$$

where $AI_l$ is the absolute importance of $CR_l$ based on the KANO model, $w_l^{LCI}$ is the loss cost importance of $CR_l$ based on the KANO model, $o_l^{MR}$, $o_l^{OR}$, $o_l^{AR}$, $o_l^{IR}$ and $o_l^{RR}$ are the numbers of $CR_l$ categorized in the above five requirements categories in the valid questionnaires.

**Step 3:** Calculation of the quality level increase rate. The service design team performs a thorough evaluation and rating of the current market, focusing on two aspects: (1) the degree to which the company's products satisfy each customer requirement level; and (2) the degree to which competitors' products satisfy the same customer requirement level. The service design team then analyzes the company's target level to be achieved for each customer requirement and thus obtains the quality level increase rate:

$$w_l^P = \frac{p_l{'}}{p_l} \tag{5}$$

**Table 3. The numerical criteria of customer requirement importance in the KANO model.**

| requirement categories | Attractive(AR) | One-dimensional(OR) | Must-be(MR) | Indifferent(IR) | Reverse(RR) |
|---|---|---|---|---|---|
| Numerical criteria | 1 | 3 | 5 | 0 | -3 |

where $w_l^P$ is the quality level increase rate of $CR_l$ based on the existing market competitiveness evaluation, $p_l$ is the current quality level of $CR_l$, and $p_l$' is the target quality level of $CR_l$.

**Step 4:** Determination of the importance weight of customer requirements. Based on the above customer requirement importance judgment matrix, KANO model, and competitiveness evaluation analysis, the relative importance weight of customer requirements can be determined. The formula for relative importance weight is as follows:

$$w_l = \frac{w_l'}{\sum_{l=1}^{d} w_l'} \times 100\% \tag{6}$$

$$w_l' = w_l^Z \times w_l^{LCI} \times w_l^P \tag{7}$$

where $w_l$ is the final relative importance weight of $CR_l$.

## 4.2. Calculation of service module utility value when satisfying different requirements

Influenced by its own properties, the utility values of different candidate items $SC_{ij}^l$ under each service module will have different trends as their parameter values $b_{ij}^l$ change, that is, the utility measurement functions of different candidate itemsets $SC_i^l$ under each service module will be different. Thus, before measuring the utility of the candidate items, it is necessary to determine the appropriate candidate item utility measurement function according to the properties of the candidate items, and on this basis, calculate the candidate item utility value, and then obtain the utility value of each service module, as follows:

First of all, combined with the previous case study conducted by scholars [38], the utility measure function design schemes constructed for the candidate items in each situation are as follows:

(1) Benefit-form binary utility measure function

As shown in the schematic diagram in Fig 5, the candidate item decision's utility structure follows the "0-1" binary format. This means that if the parameter value $b_{ij}^l$ of the selected candidate item equals or exceeds the expected level (in numerical or set form) under its correlating customer requirement, the utility value is 1; otherwise, it is 0. Therefore, the benefit-form binary utility measure function of the candidate decision is defined as follows:

$$f^1(b_{ij}^l) = \begin{cases} 0, & b_{ij}^l < g_1 \\ 1, & b_{ij}^l \geq g_1 \end{cases} \tag{8}$$

$$f^2(b_{ij}^l) = \begin{cases} 0, & b_{ij}^l \not\subset G \\ 1, & b_{ij}^l \subseteq G \end{cases} \tag{9}$$

where $f^1(b_{ij}^l)$ and $f^2(b_{ij}^l)$ are the utility values when the expected level of $b_{ij}^l$ is in numerical and set form, respectively; $b_{ij}^l$ is the parameter value of candidate item $SC_{ij}^l$; $g_1$ and $G$ are the expected levels of $b_{ij}^l$ under the correlating customer requirements in numerical and set form, respectively.

(2) Benefit-form ladder utility measure function

As shown in the schematic diagram in Fig 6, the candidate item decision's utility structure follows the "0- $\alpha_1$ - $\alpha_2$ -1" ladder format. This means that if the parameter value $b_{ij}^l$ of the

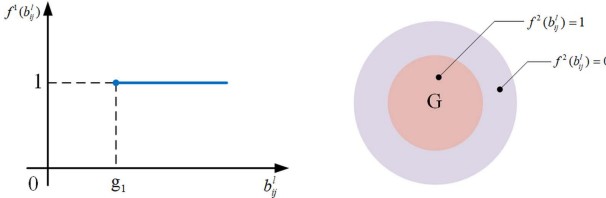

**Fig 5. The schematic diagram of benefit-form binary utility measure functions** $f^1(b^l_{ij})$ **and.** $f^2(b^l_{ij})$.

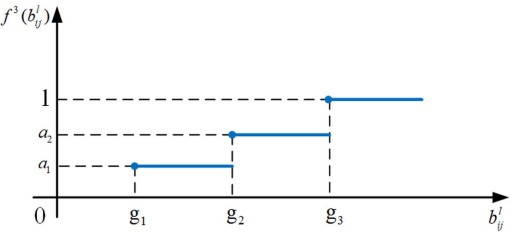

**Fig 6. The schematic diagram of the benefit-form ladder utility measure function.** $f^3(b^l_{ij})$.

selected candidate item is less than its lower limit under the correlating customer requirement, the utility value is 0; If $b^l_{ij}$ is in the interval $[g_1, g_2)$, the utility value is $\alpha_1$; If $b^l_{ij}$ is in the interval $[g_2, g_3)$, the utility value is $\alpha_2$, otherwise, it is 1. Therefore, the benefit-form ladder utility measure function of the candidate decision is defined as follows:

$$f^3(b^l_{ij}) = \begin{cases} 0, & b^l_{ij} < g_1 \\ \alpha_1, & g_1 \leq b^l_{ij} < g_2 \\ \alpha_2, & g_2 \leq b^l_{ij} < g_3 \\ 1, & b^l_{ij} \geq g_3 \end{cases} \tag{10}$$

where $f^3(b^l_{ij})$ is the utility value of the corresponding candidate item decision; $b^l_{ij}$ is the parameter value of $SC^l_{ij}$; $g_1$, $g_2$, and $g_3$ are the boundary values of $b^l_{ij}$; $\alpha_1$ and $\alpha_2$ are the utility values corresponding to $b^l_{ij}$ in different intervals, $0 < \alpha_1 < \alpha_2 < 1$, $g_1 < g_2$.

(3) Benefit-form parabola utility measure function

As shown in the schematic diagram in Fig 7, the candidate item decision's utility structure follows the $[0,1]$ parabolic growth format. This means that if the parameter value $b^l_{ij}$ of the selected candidate item is less than its lower limit under the correlating customer requirement, the utility value is 0; If $b^l_{ij}$ is in the interval $[g_1, g_2)$, utility value will increase in the form of diminishing marginal utility and the growth range is $[0,1)$; otherwise, it is 1. Therefore, the benefit-form parabola utility measure function of the candidate decision is defined as follows:

$$f^4(b^l_{ij}) = \begin{cases} 0, & b^l_{ij} < g_1 \\ \sin\dfrac{\pi}{2(g_2 - g_1)}(b^l_{ij} - g_1), & g_1 \leq b^l_{ij} < g_2 \\ 1, & b^l_{ij} \geq g_2 \end{cases} \tag{11}$$

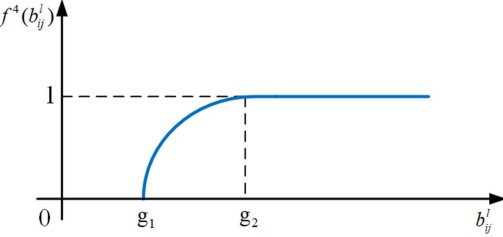

**Fig 7. The schematic diagram of the benefit-form parabola utility measure function.** $f^4(b_{ij}^l)$.

where $f^4(b_{ij}^l)$ is the utility value of the corresponding candidate item decision; $b_{ij}^l$ is the parameter value of $SC_{ij}^l$; $g_1$ and $g_2$ are the boundary values of $b_{ij}^l$, $g_1 < g_2$.

(4) Interval-form parabola utility measure function

As shown in the schematic diagram in Fig 8, the candidate item decision's utility structure follows the $[0,1]$ interval format. This means that if the parameter value $b_{ij}^l$ of the selected candidate item is less than its lower limit or more than its upper limit under the correlating customer requirement, the utility value is 0; If $b_{ij}^l$ is in the interval $[g_1,g_2)$, utility value will increase in the form of diminishing marginal utility and the growth range is $[0,1)$; If $b_{ij}^l$ is in the interval $[g_2,g_3)$, the utility value is 1; If $b_{ij}^l$ is in the interval $[g_3,g_4)$, utility value will decrease in the form of increasing marginal utility and the growth range is $(0,1]$. Therefore, the interval-form parabola utility measure function of the candidate decision is defined as follows:

$$f^5(b_{ij}^l) = \begin{cases} 0, & b_{ij}^l < g_1 \\ \sin\dfrac{\pi}{2(g_2 - g_1)}\left(b_{ij}^l - g_1\right), & g_1 \leq b_{ij}^l < g_2 \\ 1, & g_2 \leq b_{ij}^l < g_3 \\ -\sin\dfrac{\pi}{2(g_4 - g_3)}\left(b_{ij}^l - g_4\right), & g_3 \leq b_{ij}^l < g_4 \\ 0, & b_{ij}^l \geq g_4 \end{cases} \quad (12)$$

where $f^5(b_{ij}^l)$ is the utility value of the corresponding candidate item decision; $b_{ij}^l$ is the parameter value of $SC_{ij}^l$; $g_1$, $g_2$, $g_3$ and $g_4$ are the boundary values of $b_{ij}^l$, $g_1 < g_2 < g_3 < g_4$.

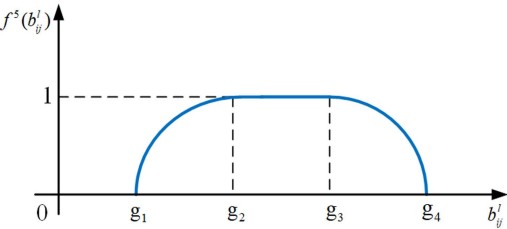

**Fig 8. The schematic diagram of the interval-form parabola utility measure function.** $f^5(b_{ij}^l)$.

Then, calculates the utility value of service modules. This study assumes that among each available service module, only one candidate item can be selected from the candidate itemset correlated with each customer requirement to form the service module's candidate item configuration scheme and then form the final service resource configuration scheme. Therefore, once a service module selects specific service candidate items, it can calculate its utility value as follows:

$$u_i^l = \sum_{j=1}^{n_i^l} f(b_{ij}^l) x_{ij}^l \tag{13}$$

where $u_i^l$ is the utility value of $SM_i$ when satisfying the customer requirement $CR_l$; $x_{ij}^l$ is a "0-1" decision variable, if candidate items $SC_{ij}^l$ is selected, $x_{ij}^l = 1$, otherwise, $x_{ij}^l = 0$. $f(b_{ij}^l)$ is the utility function value of $SC_{ij}^l$, it should be pointed out that when calculating $f(b_{ij}^l)$, it is necessary to select the utility measure function that is suitable for $SC_{ij}^l$ from the above four types of utility measure functions and then calculate the utility function value $f(b_{ij}^l)$. The utility measure function will vary based on the candidate itemsets. In this way, the utility value of each service module can be obtained when satisfying different requirements.

## 4.3. Determination of customer satisfaction with service resource configuration schemes

In the QFD theory, the HoQ allows us to obtain the correlation matrix between customer requirements and technical attributes, as well as between technical attributes and service modules. By combining these matrices, we can derive the correlation matrix between customer requirements and service modules, as depicted in Fig 3. The details are as follows:

$R^1 = \left[r_{la}^1\right]_{d \times v}$ is the correlation matrix between customer requirements and technical attributes obtained through HoQ, where $r_{la}^1$ represents the correlation coefficient between $CR_l$ and $TA_a$, which is used to indicate the level of correlation between customer requirements and technical attributes; $l = 1, 2, ..., d$, $a = 1, 2, ..., v$. Usually, the correlation coefficients are provided directly by specialists or the service design team.

$R^2 = \left[r_{ai}^2\right]_{v \times m}$ is the correlation matrix between technical attributes and service modules obtained through HoQ, where $r_{ai}^2$ represents the correlation coefficient between $TA_a$ and $SM_i$, which is used to indicate the level of correlation between technical attributes and service modules; $a = 1, 2, ..., v$, $i = 1, 2, ..., m$. Usually, the correlation coefficients are provided directly by specialists or the service design team.

$U = \left[u_i^{l'}\right]_{m \times d}$ is the utility matrix of different service modules in a service resource configuration scheme when satisfying different customer requirements, where $u_i^{l'}$ is the utility value of $SM_i$ when satisfying the customer requirement $CR_{l'}$, $i = 1, 2, ..., m$, $l' = 1, 2, ..., d$.

The satisfaction utility matrix of the service resource configuration scheme for each customer requirement is as follows:

$$Q = R^1 \cdot R^2 \cdot U = \left[r_{la}^1\right]_{d \times v} \left[r_{ai}^2\right]_{v \times m} \left[u_{il'}\right]_{m \times d} E_d = \left[q_{ll'}\right]_{d \times d} \tag{14}$$

where $q_{ll'}$ is the satisfaction utility of the service resource configuration scheme for $CR_{l'}$ when it satisfies $CR_l$, $l' = l = 1, 2, ..., d$; $E_d$ is the identity matrix.

Then the satisfaction utility value of the service resource configuration scheme for the customer is as follows:

$$q = \sum_{l=1}^{d} w_l q_{ll'}, \qquad l = l' = 1, 2, ..., d \tag{15}$$

where $q$ is the satisfaction utility value of the service resource configuration scheme for the customer. Based on the existing constraints, the objective of this study is to select the service resource configuration scheme with the highest satisfaction utility value among the generated schemes.

## 4.4. Cost calculation of service resource configuration scheme

To calculate the cost of the service resource configuration scheme, we first need to establish a cost set of feasible candidate item configuration schemes under each service module in the service resource configuration scheme. Then, we can compare the obtained candidate item configuration schemes with the cost set to obtain the cost of each service module and then integrate the total cost of the service resource configuration scheme, as shown in Fig 2. The specific process is as follows:

$C_i = \{c_{i1}, c_{i2}, ..., c_{it^i}\}$ is cost set of $MCS_i^*$, where $c_{is}$ represents the cost of $OC_{is}^*$, $i = 1, 2, ..., m$, $s = 1, 2, ..., t^i$.

It is known that $OC_{is}^*$ is the final candidate item configuration scheme for the service module $SM_i$. At this time, the cost of the service module $c_i$ can be determined based on the cost set $C_i$ and the determined candidate item configuration scheme $OC_{is}^*$, that is, $c_i = c_{is}$, $i = 1, 2, ..., m$, $s = 1, 2, ..., t^i$.

The cost calculation formula for the service resource configuration scheme is as follows:

$$c_{sum} = c_{fix} + \sum_{i=1}^{m} c_i \tag{16}$$

where $c_{sum}$ is the sum cost of the service resource configuration scheme, $c_{fix}$ is the fixed costs incurred during the design, production, and installation of the service resource configuration scheme, $c_i$ is the cost of the service module $SM_i$.

## 4.5. Construction of mathematical model

$$Max \ q = \sum_{l=1}^{d} \sum_{i=1}^{m} \sum_{j=1}^{n_i^l} w_l q_{ll} \cdot x_{ij}^l \tag{17}$$

$$\sum_{SC_{ij}^{l'} \in SC_i^{l'}} x_{ij}^l = 1 \tag{18}$$

$$l' = l, l = 1, 2, ..., d \tag{19}$$

$$c_{sum} \leq c_{up} \tag{20}$$

$$x_{kij} = 0 \ or \ 1, i = 1, 2, ..., m; j = 1, 2, ..., n, k = 1, 2, ..., h \tag{21}$$

$$\prod_{\substack{SC_{ij}^l \in OC_{ik} \\ SC_{i^*j^*}^j \in OC_{is}^*}} y_{kij(si^*j^*)}^l = 1 \tag{22}$$

In this mathematical model, formula (17) is the objective function, which means maximizing the satisfaction utility value of the service resource configuration scheme when satisfying customer requirements; Formula (18) represents that only one candidate item can be selected from each candidate itemset; Formula (19) ensures that only candidate items that satisfy the

correlating customer requirements can be selected; Formula (20) represents that the sum cost of the service resource configuration schemes cannot exceed the specified upper limit; Formula (21) represents that the decision variable can only take the value of 0 or 1; Formula (22) represents that the candidate item configuration scheme used for configuration must be feasible.

## 5. Solution using IGA algorithm

When dealing with large-scale problems, Exact Algorithms (EA) may encounter a range of issues, including combinatorial explosion, lengthy optimization time, and failure to reach the optimal solution within a reasonable timeframe. Genetic Algorithm (GA), as an effective meta-heuristic algorithm, has been widely used in various engineering optimization problems in product development due to its advantages such as rapid search speed, strong robustness, and excellent convergence effect. Hence, to address the service resource configuration optimization problem in this study, an improved genetic algorithm (IGA) was devised. This algorithm takes into account the candidate item selection problem and the cost screening and matching problem of the service module, considering them from the perspective of multi-requirement correlation. The specific process of the IGA is depicted in Fig 9.

**Step 1:** Encoding. The initial stage involves encoding the problem that needs to be resolved. Encoding involves the transformation of feasible solutions from the solution space of a problem to the search space of IGA. This is the initial stage of IGA. It is necessary to ascertain the solution's interval and precision and thereafter encode the initial population based on these parameters. There are several methods of encoding, such as binary encoding, grey code, real number encoding, and character encoding.

**Step 2:** Population initialization. Before commencing the genetic algorithm iteration process, it is necessary to initialize the population and establish parameters such as population size, population number, DNA length, number of evolutions, and mutation rate. The population's variety is defined by its size, whereas the encoding process defines the length of the chromosome. The initial population is often generated randomly. However, if the actual distribution of the population is known, the initial population can also be formed based on this distribution.

**Step 3:** Fitness value calculation and selection operation. Calculate the fitness value of each individual, which represents their performance in the problem space. The fitness function is a criterion for measuring the quality of individuals, which is defined according to the specific objective of the optimization problem; the selection operation is the process of selecting individuals from the previous generation population to the next generation population. Generally, individuals are selected based on the distribution of individual fitness.

**Step 4:** Criss-cross inheritance operation. The criss-cross inheritance operation involves the random selection of two individuals from a group of selected individuals. Their gene sequences are subsequently exchanged at one or more specific places to generate new progeny. This method emulates genetic recombination in biological reproduction.

**Step 5:** Mutation Operation. The mutation operation is to make a small probability random change to the gene sequence of the newly generated individuals in order to increase the population's diversity. This helps the algorithm jump out of the local optimal solution and explore a wider solution space.

**Step 6:** Gene sequence matching and new population evaluation. In contrast to traditional genetic algorithms (GA), considering the inconsistency of candidate itemsets of service modules from the perspective of multi-requirement correlation, this study adds a gene sequence matching process when using genetic algorithms to ensure that the generated individuals actually exist, and then calculates the fitness values of the new individuals generated after the

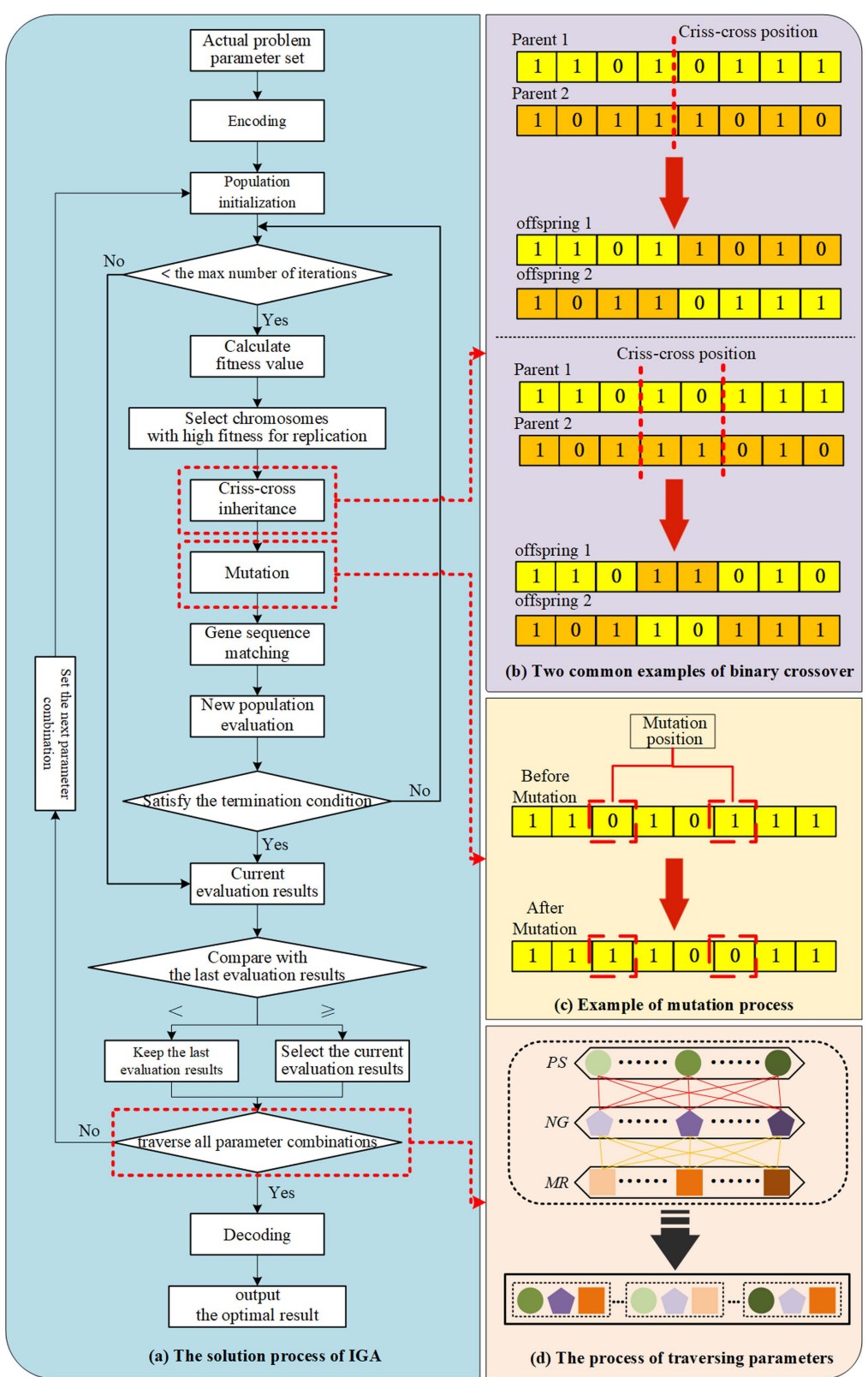

**Fig 9.** The specific solution process of the IGA.

operation. The specific screening and matching process is consistent with the yellow background color labeling part in Fig 2.

**Step 7:** Update population. The original population is replaced with individuals of the new generation, and then the above selection, crisscross inheritance, mutation, matching, and evaluation process is repeated until the termination criterion is achieved. The termination criterion can be reaching the maximum number of iterations or the fitness value reaching the preset target threshold.

**Step 8:** Comparison of evaluation results. The fitness value of the population evaluated at the current parameter level set is compared with the fitness value obtained at the previous parameter level, and the population with a higher fitness value is retained as the comparison population. This cycle is iterated until all given parameter combinations are traversed. This process is the outer loop cycle process in Fig 9.

**Step 9:** Decoding and outputting results. Decode the encoded chromosome to obtain the original form of the problem solution. Then, output the current optimal solution and its corresponding ideal fitness value.

## 6. Case study

To validate the efficacy and viability of the method proposed in this study, we demonstrate its potential applicability in the field of living room customization.

Within the whole-home customization industry, as customer requirements become more sophisticated and diversified, the requirements for the diversity of decoration resources and the flexibility of decoration processes are gradually increasing. To provide high-quality customized services, a large whole-home customization company intends to improve the whole-home customization service products of houses by taking living room customization as an example.

In the whole customization process, modules can be divided based on indoor space, such as windows, floors, ceilings, etc., which can all be used as a single service module. A complete whole customization service resource configuration scheme can be obtained by selecting and configuring all modules. In this process, customers' requirements need to be met by related service modules, but the candidate itemsets that need to be configured for the same service module will be different when meeting different requirements. For example, the air conditioner pays attention to the configuration of its temperature control system when meeting the temperature requirement of customers, and pays attention to the configuration of its humidity control system when meeting the humidity requirement of customers.

The company comprehensively determined the six customer requirements for living room customization in the whole-home customization process through the distribution of questionnaires, website browsing, expert interviews, and targeted communication and investigation with customers, as shown in Table 4, and defined eight technical attributes, as shown in Table 5.

Constructing a modular framework for living room customization. The living room customization process is divided into 8 service modules. Simultaneously, combined with the actual situation, a service module may satisfy multiple customer requirements. The candidate itemsets in the same service module will be different due to the different customer requirements satisfied by the service module. Therefore, several candidate itemsets are established for each service module, and several candidate items are established for each candidate itemset. Each candidate item in the same candidate itemset has similar functions and effects and can be replaced by each other, but there will be certain differences in performance, cost, and quality. Table 6 below illustrates the hierarchical division of service modules and candidate vTo address the problem of service resource configuration optimization in living room

**Table 4. Customer requirements.**

| Name | Specific segmentation |
|---|---|
| Customer requirements $CR$ | $CR_1$ : Suitable temperature, warm in winter and cool in summer |
| | $CR_2$ : Suitable humidity, waterproof and moisture-proof |
| | $CR_3$ : Lighting to meet different requirements |
| | $CR_4$ : Environmental friendly |
| | $CR_5$ : Energy saving |
| | $CR_6$ : Good sound insulation |

**Table 5. Technical attributes.**

| Name | Specific segmentation |
|---|---|
| Technical attributes $TA$ | $TA_1$ : Temperature control |
| | $TA_2$ : Humidity control |
| | $TA_3$ : Waterproof design |
| | $TA_4$ : Soundproofing design |
| | $TA_5$ : Lighting adjustment |
| | $TA_6$ : Daylighting and ventilation |
| | $TA_7$ : Environmental performance |
| | $TA_8$ : Energy-saving design |

**Table 6. The hierarchical division of service modules and candidate itemsets.**

| service module | candidate itemset | candidate itemset interpretation |
|---|---|---|
| Air Conditioner selection ($SM_1$) | $SC_1^1 = \left\{ SC_{11}^1, SC_{12}^1, ..., SC_{17}^1 \right\}$ | HP = {WM-1hp,WM-1.5hp,V-1.5hp,V-2hp,V-3hp,C-3hp,C-4hp} |
| | $SC_1^2 = \left\{ SC_{11}^2 \right\}$ | Adjust humidity range = {65%} |
| | $SC_1^5 = \left\{ SC_{11}^5, SC_{12}^5, SC_{13}^5 \right\}$ | energy efficiency index = {Level 1-4.78, Level 2-4.19, Level 3-3.99} |
| Heating equipment selection ($SM_2$) | $SC_2^1 = \left\{ SC_{21}^1, SC_{22}^1, ..., SC_{2(10)}^1 \right\}$ | Watt = {E-2000W,E-2100W,E-2200W,E-3000W,R-1800W,R-1900W, R-2000W,R-2100W,R-2200W,R-2440W} |
| Window selection ($SM_3$) | $SC_3^{123} = \left\{ SC_{31}^{123}, SC_{32}^{123}, ..., SC_{35}^{123} \right\}$ | Window area = {4.5 $m^2$ ,5 $m^2$ ,5.5 $m^2$ ,6 $m^2$ ,7 $m^2$ } |
| | $SC_3^6 = \left\{ SC_{31}^6, SC_{32}^6, ..., SC_{36}^6 \right\}$ | Sound insulation decibel = {25dB,26dB,27dB,28dB,29dB,30dB} |
| Flooring selection ($SM_4$) | $SC_4^2 = \left\{ SC_{41}^2, SC_{42}^2, SC_{43}^2 \right\}$ | Floor material = {marble, Strengthening compound, Solid wood composite} |
| Interior paint selection ($SM_5$) | $SC_5^4 = \left\{ SC_{51}^4, SC_{52}^4, SC_{53}^4 \right\}$ | Environmental protection level = {France-A +,USA-GREENGUARD, ACEM-10} |
| Lamps and lighting design ($SM_6$) | $SC_6^3 = \left\{ SC_{61}^3, SC_{62}^3, ..., SC_{66}^3 \right\}$ | Illuminance = {200lx,280lx,330lx,350lx,450lx,500lx} |
| | $SC_6^5 = \left\{ SC_{61}^5, SC_{62}^5, ..., SC_{65}^5 \right\}$ | Luminous efficacy = {70lm/W,90 lm/W,100 lm/W,110 lm/W,130 lm/W} |
| Ceiling material selection ($SM_7$) | $SC_7^4 = \left\{ SC_{71}^4, SC_{72}^4, SC_{73}^4 \right\}$ | Ceiling materials = {PVC, Aluminum Alloy, Gypsum Board} |
| | $SC_7^6 = \left\{ SC_{71}^6, SC_{72}^6, ..., SC_{74}^6 \right\}$ | Sound insulation decibel = {46dB,47dB,48dB,49dB} |
| Cabinet customization ($SM_8$) | $SC_8^2 = \left\{ SC_{81}^2, SC_{82}^2, SC_{83}^2 \right\}$ | Plate = {Particle board, Plywood, Ecological board} |
| | $SC_8^4 = \left\{ SC_{81}^4, SC_{82}^4, SC_{83}^4 \right\}$ | Environmental protection level = {ENF,E0,E1} |
| Color temperature adjustment ($SM_9$) | $SC_9^3 = \left\{ SC_{91}^3, SC_{92}^3 \right\}$ | Dimming method = {Three-color dimming mode, Stepless dimming mode} |

customization, the following is a concise overview of the research process that utilizes the service resource configuration optimization method above to determine the service resource configuration scheme.

It is known that the living room studied in this example covers an area of 30 square meters and is located in Shenyang City, Liaoning Province, China. The budget cost is 25,000 yuan. The correlation between the "customer requirements-technical attributes-service modules-candidate itemsets" relationship is shown in Table 7.

The cost of each candidate item configuration scheme in the existing service module (product) in this case is shown in Table 8 below. $c_{fix} = 4500$ RMB.

## 6.1. Quantification of the relative importance weight of customer requirements

Calculate the relative importance weight of customer requirements. The importance judgment matrix, KANO model, and competitiveness evaluation are integrated to evaluate the relative importance weight of customer requirements. We investigated 148 adult customers, not involving minors, and distributed questionnaires to them, mainly to investigate their requirements for the decoration industry and give their importance judgments in May-June 2024. Participants were provided with the right to information and verbal consent was given by the participant, which was witnessed with a colleague who was investigating with them.

In order to ensure the rationality, validity and authenticity of the obtained data, and ensure that it can fully reflect the requirements of customers in Shenyang for living room

**Table 7. The correlation between the "$CR$ - $TA$ - $SM$ - $SC_i^l$".**

| $CR$ | $TA$ | $SM$ | $SC_i^l$ |
|---|---|---|---|
| $CR_1$ | $TA_1$ | $SM_1$ | $SC_1^1 = \left\{ SC_{11}^1, SC_{12}^1, ..., SC_{17}^1 \right\}$ |
| | | $SM_2$ | $SC_2^1 = \left\{ SC_{21}^1, SC_{22}^1, ..., SC_{2\blacklozenge10\blacklozenge}^1 \right\}$ |
| | $TA_6$ | $SM_3$ | $SC_3^1 = \left\{ SC_{31}^1, SC_{32}^1, ..., SC_{35}^1 \right\}$ |
| $CR_2$ | $TA_3$ | $SM_4$ | $SC_4^2 = \left\{ SC_{41}^2, SC_{42}^2, SC_{43}^2 \right\}$ |
| | | $SM_8$ | $SC_8^2 = \left\{ SC_{81}^2, SC_{82}^2, SC_{83}^2 \right\}$ |
| | $TA_2$ | $SM_1$ | $SC_1^2 = \left\{ SC_{11}^2 \right\}$ |
| | $TA_6$ | $SM_3$ | $SC_3^2 = \left\{ SC_{31}^2, SC_{32}^2, ..., SC_{35}^2 \right\}$ |
| $CR_3$ | $TA_5$ | $SM_6$ | $SC_6^3 = \left\{ SC_{61}^3, SC_{62}^3, ..., SC_{66}^3 \right\}$ |
| | | $SM_9$ | $SC_9^3 = \left\{ SC_{91}^3, SC_{92}^3 \right\}$ |
| | $TA_6$ | $SM_3$ | $SC_3^3 = \left\{ SC_{31}^3, SC_{32}^3, ..., SC_{35}^3 \right\}$ |
| $CR_4$ | $TA_7$ | $SM_5$ | $SC_5^4 = \left\{ SC_{51}^4, SC_{52}^4, SC_{53}^4 \right\}$ |
| | | $SM_7$ | $SC_7^4 = \left\{ SC_{71}^4, SC_{72}^4, SC_{73}^4 \right\}$ |
| | | $SM_8$ | $SC_8^4 = \left\{ SC_{81}^4, SC_{82}^4, SC_{83}^4 \right\}$ |
| $CR_5$ | $TA_8$ | $SM_1$ | $SC_1^5 = \left\{ SC_{11}^5, SC_{12}^5, SC_{13}^5 \right\}$ |
| | | $SM_6$ | $SC_6^5 = \left\{ SC_{61}^5, SC_{62}^5, ..., SC_{65}^5 \right\}$ |
| $CR_6$ | $TA_4$ | $SM_3$ | $SC_3^6 = \left\{ SC_{31}^6, SC_{32}^6, ..., SC_{36}^6 \right\}$ |
| | | $SM_7$ | $SC_7^6 = \left\{ SC_{71}^6, SC_{72}^6, ..., SC_{74}^6 \right\}$ |

**Table 8. The cost of each candidate item configuration scheme in the existing service module.**

| Service modules | Feasible candidate item configuration scheme | Corresponding cost |
|---|---|---|
| $SM_1$ | $OC_{11}^* = \{SC_{11}^1, SC_{11}^2, SC_{11}^5\}$, $OC_{12}^* = \{SC_{12}^1, SC_{11}^2, SC_{11}^5\}$, $OC_{13}^* = \{SC_{13}^1, SC_{11}^2, SC_{11}^5\}$, $OC_{14}^* = \{SC_{14}^1, SC_{11}^2, SC_{11}^5\}$, $OC_{15}^* = \{SC_{15}^1, SC_{11}^2, SC_{11}^5\}$, $OC_{16}^* = \{SC_{16}^1, SC_{11}^2, SC_{11}^5\}$, $OC_{17}^* = \{SC_{17}^1, SC_{11}^2, SC_{11}^5\}$, $OC_{18}^* = \{SC_{15}^1, SC_{11}^2, SC_{12}^5\}$, $OC_{19}^* = \{SC_{16}^1, SC_{11}^2, SC_{12}^5\}$, $OC_{1(10)}^* = \{SC_{11}^1, SC_{11}^2, SC_{13}^5\}$, $OC_{1(11)}^* = \{SC_{12}^1, SC_{11}^2, SC_{13}^5\}$, $OC_{1(12)}^* = \{SC_{13}^1, SC_{11}^2, SC_{13}^5\}$, $OC_{1(13)}^* = \{SC_{14}^1, SC_{11}^2, SC_{13}^5\}$, $OC_{1(14)}^* = \{SC_{15}^1, SC_{11}^2, SC_{13}^5\}$, $OC_{1(15)}^* = \{SC_{16}^1, SC_{11}^2, SC_{13}^5\}$. | $c_{11} = 1799$, $c_{12} = 2049$, $c_{13} = 2649$, $c_{14} = 4169$, $c_{15} = 5108$, $c_{16} = 5979$, $c_{17} = 8499$, $c_{18} = 4899$, $c_{19} = 5799$, $c_{1(10)} = 1599$, $c_{1(11)} = 1844$, $c_{1(12)} = 1799$, $c_{1(13)} = 3669$, $c_{1(14)} = 4669$, $c_{1(15)} = 5589$. |
| $SM_2$ | $OC_{21}^* = \{SC_{21}^1\}$, $OC_{22}^* = \{SC_{22}^1\}$, $OC_{23}^* = \{SC_{23}^1\}$, $OC_{24}^* = \{SC_{24}^1\}$, $OC_{25}^* = \{SC_{25}^1\}$, $OC_{26}^* = \{SC_{26}^1\}$, $OC_{27}^* = \{SC_{27}^1\}$, $OC_{28}^* = \{SC_{28}^1\}$, $OC_{29}^* = \{SC_{29}^1\}$, $OC_{2(10)}^* = \{SC_{2(10)}^1\}$. | $c_{21} = 219$, $c_{22} = 399$, $c_{23} = 499$, $c_{24} = 549$, $c_{25} = 879$, $c_{26} = 929$, $c_{27} = 979$, $c_{28} = 1029$, $c_{29} = 1075$, $c_{2(10)} = 1192$. |
| $SM_3$ | $OC_{3s}^* = \{SC_{3j^{123}}^{123}, SC_{3j^6}^6\}$ | $c_{3s} = area \times unit\ price$ |
| $SM_4$ | $OC_{41}^* = \{SC_{41}^2\}$, $OC_{42}^* = \{SC_{42}^2\}$, $OC_{43}^* = \{SC_{43}^2\}$. | With a loss rate of 5%, a total of 31.5 $m^2$ are required, and the cost is determined accordingly. $c_{41} = 4410$, $c_{42} = 3433.5$, $c_{43} = 3150$. |
| $SM_5$ | $OC_{51}^* = \{SC_{51}^4\}$, $OC_{51}^* = \{SC_{52}^4\}$, $OC_{53}^* = \{SC_{53}^4\}$. | The capacity of each barrel is 5 liters, and the required quantity is 5 barrels. Therefore, the cost is determined accordingly. $c_{51} = 1395$, $c_{52} = 930$, $c_{53} = 690$. |
| $SM_6$、 $SM_9$ | $OC_{61}^* = \{SC_{66}^3, SC_{65}^5, SC_{92}^3\}$, $OC_{62}^* = \{SC_{65}^3, SC_{64}^5, SC_{92}^3\}$, $OC_{63}^* = \{SC_{64}^3, SC_{63}^5, SC_{91}^3\}$, $OC_{64}^* = \{SC_{63}^3, SC_{62}^5, SC_{91}^3\}$, $OC_{65}^* = \{SC_{62}^3, SC_{62}^5, SC_{91}^3\}$, $OC_{66}^* = \{SC_{61}^3, SC_{61}^5, SC_{91}^3\}$. | $c_{61} = 719$, $c_{62} = 699$, $c_{63} = 649$, $c_{64} = 433$, $c_{65} = 527$, $c_{66} = 369$. |
| $SM_7$ | $OC_{71}^* = \{SC_{71}^4, SC_{71}^6\}$, $OC_{72}^* = \{SC_{72}^4, SC_{71}^6\}$, $OC_{73}^* = \{SC_{72}^4, SC_{74}^6\}$, $OC_{74}^* = \{SC_{73}^4, SC_{72}^6\}$, $OC_{75}^* = \{SC_{73}^4, SC_{74}^6\}$. | With a loss rate of 10%, a total of 30.3 $m^2$ are required, and the cost is determined accordingly. $c_{71} = 1212$, $c_{72} = 3333$, $c_{73} = 5454$, $c_{74} = 3333$, $c_{75} = 4545$. |
| $SM_8$ | $OC_{81}^* = \{SC_{81}^2, SC_{81}^4\}$, $OC_{82}^* = \{SC_{81}^2, SC_{82}^4\}$, $OC_{83}^* = \{SC_{82}^2, SC_{82}^4\}$, $OC_{84}^* = \{SC_{82}^2, SC_{83}^4\}$, $OC_{85}^* = \{SC_{83}^2, SC_{81}^4\}$, $OC_{86}^* = \{SC_{83}^2, SC_{82}^4\}$, $OC_{87}^* = \{SC_{83}^2, SC_{83}^4\}$. | Approximately 10 sheets of board are expected to be utilized, and the cost will be determined accordingly. $c_{81} = 2290$, $c_{82} = 1750$, $c_{83} = 2200$, $c_{84} = 2060$, $c_{85} = 2980$, $c_{86} = 2460$, $c_{87} = 2190$ |

*In $SM_3$, $SC_{3j^{123}}^{123}$ is the window area, its unit is $m^2$; $SC_{3j^6}^6$ is the window sound insulation effect, its unit price is RMB $/m^2$, so when calculating the cost of the feasible candidate configuration item scheme $OC_{3s}^*$ in $SM_3$, the formula is: $c_{3s} = area \times unit\ price$.

customization. Among the 148 adult customers in this survey, there are mainly two customer groups: the customers who have customized the living room and the customers who are customizing the living room. Both of them have the requirement for living room customization and have a certain understanding of living room customization. These 148 adult customers were randomly selected by multi-stage sampling, stratified sampling and other sampling methods, and their ages were mainly concentrated in the young and middle-aged group, that is, the 18–59 age group. The places where they live are located in ten administrative districts and three counties in Shenyang. This survey mainly investigated their requirements for the decoration industry and give their importance judgments. In this process, these 148 adult customers mainly participated in the filling of KANO model questionnaire for calculating loss cost importance.

**Step 1:** By comparing the customer requirements collected for the living room customization, the customer requirements importance judgment matrix is obtained as follows:

$$Z = \begin{bmatrix} 1 & 2 & 3 & \frac{1}{3} & 3 & 3 \\ \frac{1}{2} & 1 & 2 & \frac{1}{3} & \frac{1}{2} & 1 \\ \frac{1}{3} & \frac{1}{2} & 1 & \frac{1}{3} & 2 & 1 \\ 3 & 3 & 3 & 1 & 3 & 3 \\ \frac{1}{3} & 2 & \frac{1}{2} & \frac{1}{3} & 1 & 2 \\ \frac{1}{3} & 1 & 1 & \frac{1}{3} & \frac{1}{2} & 1 \end{bmatrix}$$

Based on the importance judgment matrix and formula (2), the $\overline{w_l^Z}$ of $CR_1$-$CR_6$ are 1.6189, 0.7418, 0.6934, 2.4980, 0.7782, and 0.6177, respectively. Based on formula (1), the relative importance of customer requirements is: $w_1^Z = 0.2319$, $w_2^Z = 0.1063$, $w_3^Z = 0.0993$, $w_4^Z = 0.3579$, $w_5^Z = 0.1115$, $w_6^Z = 0.0885$.

**Step 2:** We conducted a questionnaire survey on 148 customers within the company's jurisdiction in May-June 2024 and obtained customer requirement survey results based on the KANO model. Based on formulas (3) and (4), the $w_l^{LCI}$ based on the KANO model was determined. Table 9 displays the specific results.

**Step 3:** Calculation of the quality level increase rate. Before conducting the market competitiveness evaluation, the service design team comprehensively evaluated the characteristics and technical indicators of the entire customization industry and selected XX Company as the target competitor for analysis. The service design team utilized a five-level scale to evaluate the performance of the company and its competitors regarding each customer requirement, to determine the relative competitive advantages and disadvantages of the company and its competitors, and to recognize the areas that need to be maintained and improved in their service products. Table 10 displays the specific market competitiveness evaluation results.

**Step 4:** Based on $w_l^Z$, $w_l^{LCI}$, $w_l^P$ and formulas (6) and (7) obtained in the above process, the $w_l$ of $CR_l$ are finally calculated. Table 11 below displays the calculation results.

**Table 9. The specific results of.** $w_l^{LCI}$

| CR | AR | OR | MR | IR | RR | AI | $w_l^{LCI}$ |
|---|---|---|---|---|---|---|---|
| $CR_1$ | 27 | 47 | 74 | 0 | 0 | 538 | 0.2036 |
| $CR_2$ | 62 | 31 | 23 | 32 | 0 | 270 | 0.1022 |
| $CR_3$ | 40 | 63 | 45 | 0 | 0 | 454 | 0.1718 |
| $CR_4$ | 24 | 39 | 85 | 0 | 0 | 566 | 0.2142 |
| $CR_5$ | 17 | 87 | 28 | 16 | 0 | 418 | 0.1583 |
| $CR_6$ | 22 | 73 | 31 | 22 | 0 | 396 | 0.1499 |

**Table 10. The specific market competitiveness evaluation results.**

| CR | $p_l$ | $p_l^{'}$ | $w_l^P$ |
|---|---|---|---|
| $CR_1$ | 5 | 5 | 1.00 |
| $CR_2$ | 4 | 5 | 1.25 |
| $CR_3$ | 4 | 5 | 1.25 |
| $CR_4$ | 5 | 5 | 1.00 |
| $CR_5$ | 4 | 5 | 1.25 |
| $CR_6$ | 3 | 4 | 1.33 |

## 6.2. Determination of the "customer requirements-technical attributes-service module" correlation matrix based on QFD

Establish the correlation between customer requirements and service modules. $HoQ$ 1 establishes the correlation between customer requirements $CR$ and technical attributes $TA$, while $HoQ$ 2 establishes the correlation between technical attributes $TA$ and service modules $SM$. The correlation matrix provided by the service design team is as follows:

$$R^1 = \begin{bmatrix} 5 & 0 & 0 & 0 & 0 & 3 & 0 & 0 \\ 0 & 3 & 5 & 0 & 0 & 5 & 0 & 0 \\ 0 & 0 & 0 & 0 & 5 & 3 & 0 & 0 \\ 0 & 0 & 0 & 0 & 0 & 0 & 5 & 0 \\ 0 & 0 & 0 & 0 & 0 & 0 & 0 & 5 \\ 0 & 0 & 0 & 5 & 0 & 0 & 0 & 0 \end{bmatrix} \quad R^2 = \begin{bmatrix} 5 & 5 & 0 & 0 & 0 & 0 & 0 & 0 & 0 \\ 3 & 0 & 0 & 0 & 0 & 0 & 0 & 0 & 0 \\ 0 & 0 & 3 & 3 & 0 & 0 & 0 & 5 & 0 \\ 0 & 0 & 3 & 0 & 0 & 0 & 3 & 0 & 0 \\ 0 & 0 & 0 & 0 & 0 & 5 & 0 & 0 & 5 \\ 0 & 0 & 3 & 0 & 0 & 0 & 0 & 0 & 0 \\ 0 & 0 & 0 & 0 & 5 & 0 & 3 & 3 & 0 \\ 5 & 0 & 0 & 0 & 0 & 3 & 0 & 0 & 0 \end{bmatrix}$$

## 6.3 Construction of utility measure function for service candidate items

Table 12 displays the utility measure function of the candidate itemsets under each service module.

In Table 12, the utility measure function type of each candidate itemset is selected in Section 4.2 according to the changing trend of customer satisfaction utility with the parameters of the set. For example, the candidate item "horsepower (HP) of air conditioners" affects whether the air conditioner has sufficient cooling and heating capacity, that is, whether the

**Table 11. The calculation results of.** $w_l$

| CR | $w_l^Z$ | $w_l^{LCI}$ | $w_l^P$ | $w_l^{'}$ | $w_l$ |
|---|---|---|---|---|---|
| $CR_1$ | 0.2319 | 0.2036 | 1.00 | 4.7214 | 0.24 |
| $CR_2$ | 0.1063 | 0.1022 | 1.25 | 1.3580 | 0.07 |
| $CR_3$ | 0.0993 | 0.1718 | 1.25 | 2.1325 | 0.11 |
| $CR_4$ | 0.3579 | 0.2142 | 1.00 | 7.6662 | 0.38 |
| $CR_5$ | 0.1115 | 0.1583 | 1.25 | 2.2063 | 0.11 |
| $CR_6$ | 0.0885 | 0.1499 | 1.33 | 1.7644 | 0.09 |

**Table 12. The utility measure function of the candidate itemsets.**

| $SM$ | $SC_i^l$ | utility measure functions |
|---|---|---|
| $SM_1$ | $SC_1^1$ | $f^1(b_{1j}^1) = \begin{cases} 0, & b_{1j}^1 < 2 \\ 1, & b_{1j}^1 \geq 2 \end{cases}$ |
| | $SC_1^2$ | $f^4(b_{1j}^2) = \begin{cases} 0, & b_{1j}^2 < 30\% \\ \sin\dfrac{\pi}{2(70\%-30\%)}(b_{1j}^2-30\%) & 30\% \leq b_{1j}^2 < 70\% \\ 1 & b_{1j}^2 \geq 70\% \end{cases}$ |
| | $SC_1^5$ | $f^4(b_{1j}^5) = \begin{cases} 0, & b_{1j}^5 < 3.50 \\ \sin\dfrac{\pi}{2(4.50-3.50)}(b_{1j}^5-3.50) & 3.50 \leq b_{1j}^5 < 4.50 \\ 1 & b_{1j}^5 \geq 4.50 \end{cases}$ |
| $SM_2$ | $SC_2^1$ | $f^4(b_{2j}^1) = \begin{cases} 0, & b_{2j}^1 < 1800 \\ \sin\dfrac{\pi}{2(2200-1800)}(b_{2j}^1-1800) & 1800 \leq b_{2j}^1 < 2200 \\ 1 & b_{2j}^1 \geq 2200 \end{cases}$ |
| $SM_3$ | $SC_3^1$ $SC_3^2$ $SC_3^3$ | $f^5(b_{3j}^1) = f^5(b_{3j}^2) = f^5(b_{3j}^3) = \begin{cases} 0, & b_{3j}^1 < 4.3 \\ \sin\dfrac{\pi}{2(5-4.3)}(b_{3j}^1-4.3), & 4.3 \leq b_{3j}^1 < 5 \\ 1, & 5 \leq b_{3j}^1 < 7.5 \\ -\sin\dfrac{\pi}{2(9.8-7.5)}(b_{3j}^1-9.8), & 7.5 \leq b_{3j}^1 < 9.8 \\ 0, & b_{3j}^1 \geq 9.8 \end{cases}$ |
| | $SC_3^6$ | $f^4(b_{3j}^6) = \begin{cases} 0, & b_{3j}^6 < 25 \\ \sin\dfrac{\pi}{2(30-25)}(b_{3j}^6-25) & 25 \leq b_{3j}^6 < 30 \\ 1 & b_{3j}^6 \geq 30 \end{cases}$ |
| $SM_4$ | $SC_4^2$ | $f^3(b_{4j}^2) = \begin{cases} 0.6, & b_{4j}^2 = \text{Solid wood composite} \\ 0.8, & b_{4j}^2 = \text{Strengthening compound} \\ 1, & b_{4j}^2 = \text{marble} \end{cases}$ |
| $SM_5$ | $SC_5^4$ | $f^3(b_{5j}^4) = \begin{cases} 0, & b_{5j}^4 = \text{Third-party appraisal agency} \\ 0.6, & b_{5j}^4 = ACEM:(10) \\ 0.8, & b_{5j}^4 = USA:GREENGUARD \\ 1, & b_{5j}^4 = France:A+ \end{cases}$ |
| $SM_6$ | $SC_6^3$ | $f^5(b_{6j}^3) = \begin{cases} 0, & b_{6j}^3 < 100 \\ \sin\dfrac{\pi}{2(300-100)}(b_{6j}^3-100), & 100 \leq b_{6j}^3 < 300 \\ 1, & 300 \leq b_{6j}^3 < 500 \\ -\sin\dfrac{\pi}{2(1000-500)}(b_{6j}^3-1000), & 500 \leq b_{6j}^3 < 1000 \\ 0, & b_{6j}^3 \geq 1000 \end{cases}$ |
| | $SC_6^5$ | $f^4(b_{6j}^5) = \begin{cases} 0, & b_{6j}^5 < 50 \\ \sin\dfrac{\pi}{2(130-50)}(b_{6j}^5-50) & 50 \leq b_{6j}^5 < 130 \\ 1 & b_{6j}^5 \geq 130 \end{cases}$ |

*(Continued)*

**Table 12.** (Continued)

| $SM$ | $SC_i^l$ | utility measure functions |
|---|---|---|
| $SM_7$ | $SC_7^4$ | $f^3(b_{7j}^4) = \begin{cases} 0.6, & b_{7j}^4 = \text{PVC} \\ 0.8, & b_{7j}^4 = \text{Aluminum Alloy} \\ 1, & b_{7j}^4 = \text{Gypsum Board} \end{cases}$ |
| | $SC_7^6$ | $f^4(b_{7j}^6) = \begin{cases} 0, & b_{7j}^6 < 45 \\ \sin \dfrac{\pi}{2(50-45)}(b_{7j}^6 - 45) & 45 \text{ d } b_{7j}^6 < 50 \\ 1 & b_{7j}^6 \text{ e } 50 \end{cases}$ |
| $SM_8$ | $SC_8^2$ | $f^3(b_{8j}^2) = \begin{cases} 0.6, & b_{8j}^2 = \text{Particle board} \\ 0.8, & b_{8j}^2 = \text{Plywood} \\ 1, & b_{8j}^2 = \text{Ecological board} \end{cases}$ |
| | $SC_8^4$ | $f^3(b_{8j}^4) = \begin{cases} 0.6, & b_{8j}^4 = \text{E1} \\ 0.8, & b_{8j}^4 = \text{E0} \\ 1, & b_{8j}^4 = \text{ENF} \end{cases}$ |
| $SM_9$ | $SC_9^3$ | $f^2(b_{9j}^3) = \begin{cases} 0, & b_{9j}^3 \not\subset \{warm\ light, Two-color, Three-color,\ Stepless\} \\ 1, & b_{9j}^3 \subseteq \{warm\ light, Two-color, Three-color,\ Stepless\} \end{cases}$ |

air conditioner can be adjusted to a specific temperature in a specific area and environment. Combined with field research and expert opinions, the customer's response to the air conditioner's temperature adjustment is in a binary form, that is, it is satisfactory to reach a specific temperature effectively, but not satisfactory to reach it. Therefore, the utility measure function of the candidate itemset $SC_1^1$ is a Benefit-form binary utility measure function. And so on.

On this basis, the setting of parameter node values in the utility measure function of each candidate itemset mainly refers to China national and local standards, some industry standards, authoritative measurement articles in related fields, expert opinions and the change trend of "satisfaction utility value-parameter" of the candidate itemset itself. For example, the refrigeration unit of civil air conditioner is "HP", and 1 HP= 2324 W. According to the recommended standards of air conditioning industry and the climatic conditions in Shenyang, the cooling load of the case house should be 150 $m^2$ and the 30 $m^2$ living room should be 4500W, so 2 HP is regarded as the parameter nodes of $SC_1^1$ that meet the standard. In a similar way, the parameter node values of the candidate set $SC_1^5$ "Energy Efficiency Index" is determined according to the Chinese national standard GB 21455–2019, namely "Minimum allowable values of the energy efficiency and energyefficiency grades for room air conditioners", and so on.

## 6.4. Model optimization results

This model optimization was programmed in Python 3.12 and executed on a personal computer with the following configuration: CPU: 12th Gen Intel(R) Core(TM) i7-12700H 2.30 GHz, RAM:16.0 GB, Operating System: Windows 11 23H2.

Combined with the candidate itemset data given under the existing service modules provided by the enterprise, the number of selectable candidate items under each candidate set is converted into binary digits: $3+4+3+2+2+1+3+3+2+3+2+2+2+2+3+3+3 = 43$ bits. Therefore, the DNA length of the improved genetic algorithm is set to 43; in addition, in the genetic algorithm, the setting of the three parameter values of population size, number

of generations, and mutation rate usually depends on the complexity of the specific problem. This is a process that requires trial and error and adjustment, and the setting of parameters will also affect the final solution's results and efficiency. Therefore, according to the complexity of the problem in this case, this study gives different parameter setting sets in the IGA for the next step of genetic algorithm parameter optimization and model solution, as shown in Table 13.

Based on the above optional IGA parameter levels, parameter optimization is performed by cross-validation traversal parameter combinations to obtain the optimal parameter combination and the corresponding model optimization results. Table 14 presents the optimal parameter combination and model solution result of service resource configuration optimization.

Fig 10 shows the iterative optimization evaluation process under the known optimal parameter combination. The objective function value generally rises as the number of IGA iterations increases. There is a significant fluctuation before the 30th generation, and it tends to be stable between the 40th and 70th generations, but there are still slight fluctuations. After the 70th generation, the optimization result of this decision can be stably obtained.

According to the cost data in Table 8, the minimum and maximum values of the actual service configuration cost are calculated to be $c_{min} = 16824 RMB$ and $c_{max} = 39649 RMB$, respectively. Therefore, we divide the cost of this case into intervals $[c_{min}, c_{max}]$ based on the actual market situation and cost data, and set the step length to 1000 RMB. Fig 11 displays the results of the cost sensitivity analysis. It can be seen that cost has a positive impact on the fitness value (i.e., customer satisfaction utility); that is, for customers, the higher the cost budget, the better the scheme will be and the more satisfied they will be. When $c > 34000 RMB$, the impact of cost on the fitness value tends to be flat; when $c=40000 RMB$, reaches the maximum value and remains unchanged, considering the cost performance and corporate profit, $c=34000 RMB$ can be used as the maximum cost limit; and when $c < 20000 RMB$, due to the low fitness value of the customer, it is not within the consideration range, so the cost limit range can be set to $[20000, 34000]$.

## 6.5. Comparison and performance analysis

Genetic Algorithm (GA), as an effective meta-heuristic algorithm, has been widely used in various engineering optimization problems in product development due to its advantages

**Table 13. Optional parameter setting sets in the IGA.**

| Parameter name | Optional parameter | | | | |
|---|---|---|---|---|---|
| PS | 100 | 200 | | 300 | |
| NG | 50 | 100 | | 200 | |
| MR | 0.05 | 0.1 | 0.15 | 0.2 | 0.25 |
| DL | 43 | | | | |

*PS-Population size；NG-Number of generations；MR-Mutation rate；DL-DNA length.

**Table 14. The optimal parameter combination and model solution results.**

| DL | PS | NG | MR | MCS* | $q$ | $c_{sum}$ |
|---|---|---|---|---|---|---|
| 43 | 300 | 100 | 0.1 | $SC_{14}^1, SC_{11}^2, SC_{11}^5, SC_{23}^1, SC_{32}^{123}, SC_{32}^6, SC_{43}^2, SC_{51}^4,$ $SC_{63}^3, SC_{62}^5, SC_{73}^4, SC_{72}^6, SC_{83}^2, SC_{81}^4, SC_{91}^3$ | 0.9328 | 24759 |

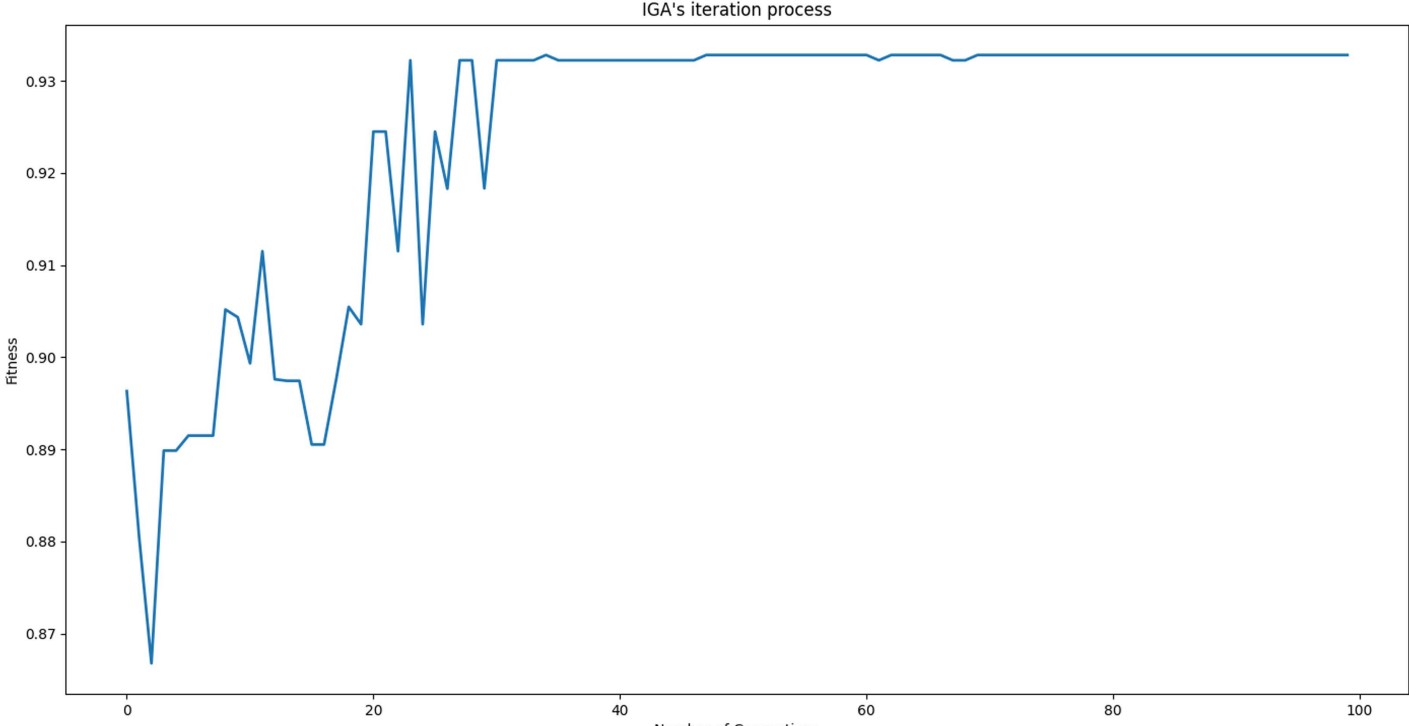

**Fig 10. The iterative optimization process based on the IGA.**

such as rapid search speed, strong robustness, and excellent convergence effect. To verify the effectiveness and superiority of the service resource configuration optimization model and the improved genetic algorithm (IGA) proposed in this paper, the IGA is compared with the original GA using the case data. Fig 11 displays the comparison results in a Box-plot.

As shown in Fig 12, this paper sets the parameters of the original GA involved in the comparison to two cases, namely GA-1: PS = 100, NG = 100, MR = 0.05, DL = 43, and GA-2: PS = 300, NG = 100, MR = 0.1, DL = 43. In the three cases, including IGA, we use Python 3.12 to run the programs 10 times, respectively, and finally compare the fitness values. It is found that the result of the IGA algorithm is better than the GA algorithm in both overall and average values. Specifically, in the 10 results, first, the lowest fitness value solved by the IGA algorithm is 0.91381596737829, which is higher than the highest fitness value solved by the GA algorithm under the two parameter combinations, namely 0.908748324334096 and 0.906090096485994; secondly, the average fitness value solved by the IGA algorithm is 0.927195658, which is 5.75% and 4.55% better than the average fitness values of GA-1 and GA-2, respectively. Finally, the results of IGA have a smaller fluctuation range and are more stable than those of GA. Therefore, this study demonstrates the superiority and positivity of the IGA algorithm in terms of overallness, average value, and stability.

## 7. Conclusion and discussion

Service resource configuration optimization is an important technical approach to achieve personalized customization and provide customers with the most suitable service. This study proposes a new service resource configuration optimization method. Compared with the available research, the primary contributions of this study are as follows:

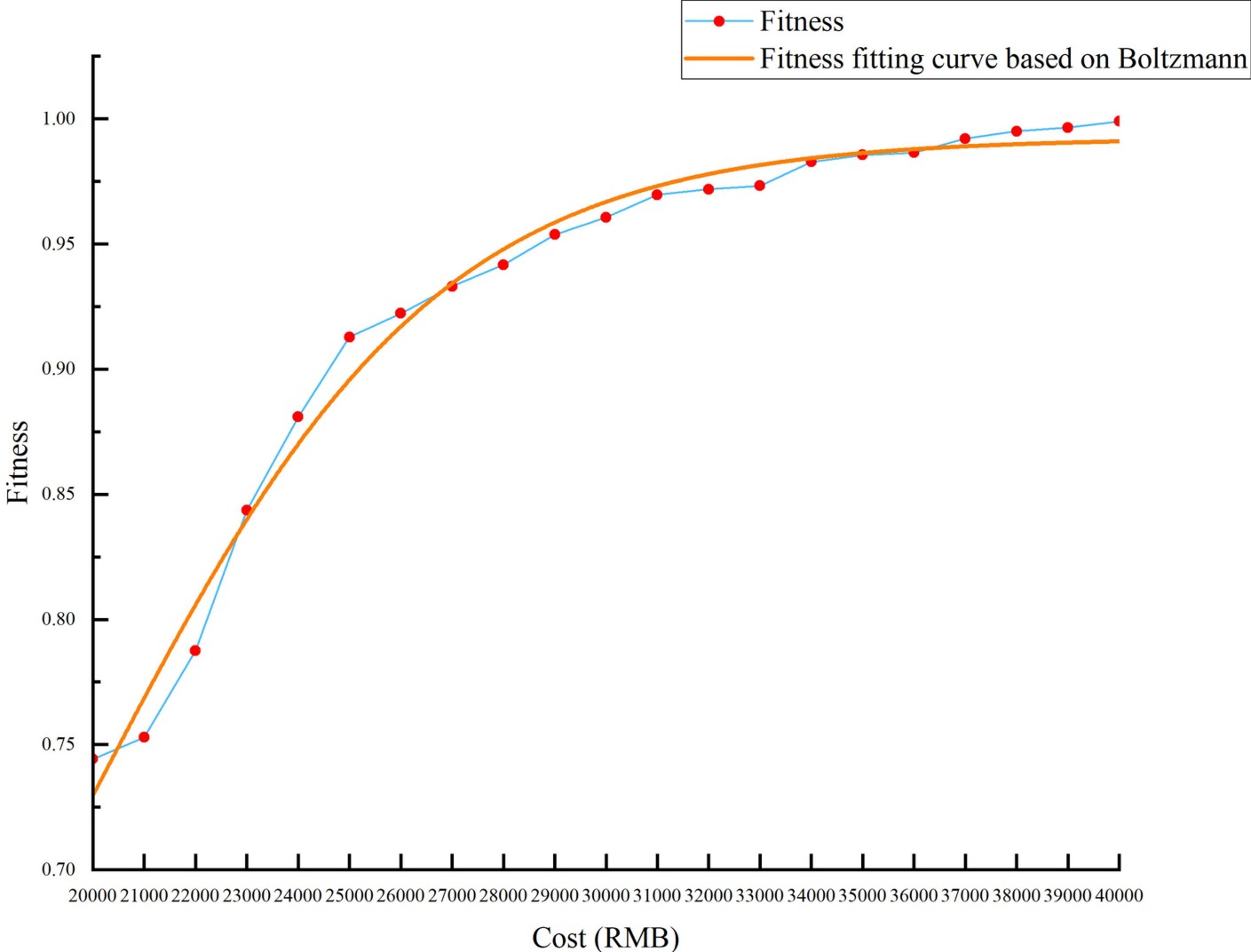

**Fig 11. The results of the cost sensitivity analysis.**

1) In the process of service requirement analysis, the importance judgment matrix, KANO model, and competitiveness evaluation are integrated to evaluate the relative importance of customer requirements, ensuring that the analysis results are more in line with the actual customer requirements;

2) In the process of service resource configuration, the "one-to-many" relationship mechanism between the service module and its candidate itemsets is considered. The candidate itemsets in the same service module will be different due to the different customer requirements satisfied by the service module, realizing the effective correlation between the service module and customer requirements;

3) An improved genetic algorithm (IGA) is designed for the problem in this study, and the effectiveness of the algorithm is proved through case analysis and algorithm comparison.

The research methods mentioned in this study can be widely extended to the customized service industry, which optimizes service resource configuration based on service

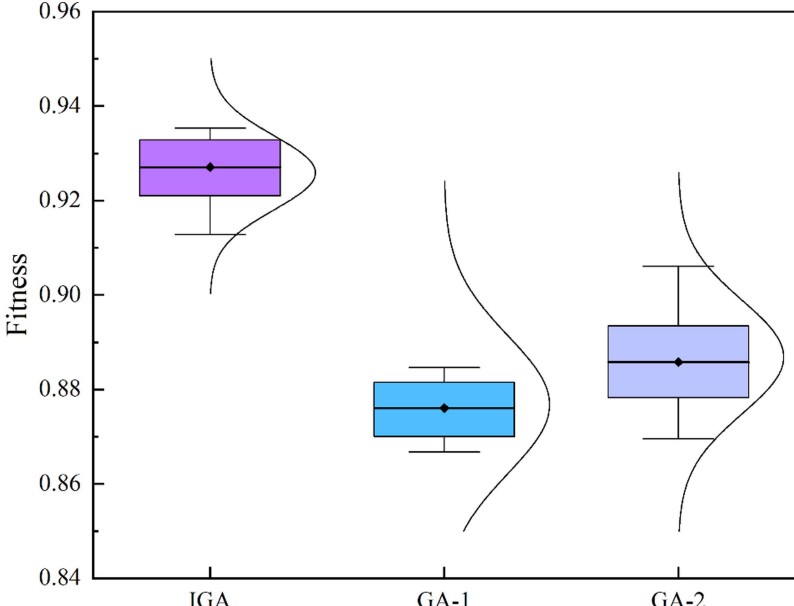

**Fig 12. The comparison result of IGA and GA.**

modularization, such as customized home appliances and software configuration in cloud environment. Taking customized smart refrigerator as an example, it is a typical modular customized product with complex structure. Smart refrigerators include intelligent coolers, compressors, thermostats and other service modules, which will have different candidate itemsets when they are associated with different customer requirements. At this time, the service requirement analysis method and the "one-to-many" relationship mechanism mentioned in this study can help enterprises make products better meet customer requirements and improve enterprise efficiency.

One limitation of this study is that it does not consider generating service resource configuration schemes for multiple customers at the same time. In practice, enterprises need to generate service resource configuration schemes for multiple customers at the same time and take into account multiple customer requirements based on the existing resources.

Another limitation is that some parameters of the optimization model proposed in this study need to be estimated beforehand, such as the QFD-based correlation matrix and the demand importance judgment matrix. In actual operation, enterprises can use their historical data to help service designers make more accurate parameter settings. In the service requirement analysis and algorithm solution, enterprises also can introduce the application of machine learning and artificial intelligence technology to improve the accuracy and effectiveness of the research results.

Future research potentials may include: 1) Based on this study, consider the situation of generating service resource configuration schemes for multiple customers at the same time, taking into account the requirements of multiple customers and the time requirements of customers, and effectively scheduling and optimizing existing resources; 2) Since customer requirements are vague, how to combine customer requirements quantitatively and qualitatively and integrate dynamic customer requirements that may change over time into service resource configuration is still a topic worthy of research; 3) From a full life cycle perspective, consider the impact of supplier behavior on platform service resource configuration, establish

a three-party interaction mechanism among suppliers, customization companies, and customers, and better help enterprises configure service resources.

## Supporting information

**S1 File. Supporting Information file.**
(DOCX)

## Author contributions

**Conceptualization:** Chao Yu.

**Data curation:** Haibin Wang.

**Funding acquisition:** Chao Yu.

**Methodology:** Haibin Wang.

**Resources:** Haibin Wang.

**Software:** Haibin Wang.

**Supervision:** Chao Yu.

**Validation:** Chao Yu.

**Writing – original draft:** Haibin Wang.

**Writing – review & editing:** Chao Yu.

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
