## [Decision Letter · Decision Letter 0]

15 Oct 2024

PONE-D-24-37417Personalized customization: service resource configuration optimization driven by customer requirements accuratelyPLOS ONE

Dear Dr. Wang,

Thank you for submitting your manuscript to PLOS ONE. After careful consideration, we feel that it has merit but does not fully meet PLOS ONE’s publication criteria as it currently stands. Therefore, we invite you to submit a revised version of the manuscript that addresses the points raised during the review process.Please submit your revised manuscript by Nov 29 2024 11:59PM. If you will need more time than this to complete your revisions, please reply to this message or contact the journal office at plosone@plos.org . Please include the following items when submitting your revised manuscript:

We look forward to receiving your revised manuscript.

Kind regards,

Anurag Sinha, Ph.D

Academic Editor

PLOS ONE

Journal Requirements:

2. You indicated that ethical approval was not necessary for your study. We understand that the framework for ethical oversight requirements for studies of this type may differ depending on the setting and we would appreciate some further clarification regarding your research. Could you please provide further details on why your study is exempt from the need for approval and confirmation from your institutional review board or research ethics committee (e.g., in the form of a letter or email correspondence) that ethics review was not necessary for this study? Please include a copy of the correspondence as an ""Other"" file.

 Funding statement: This work was supported in part by the Liaoning Provincial Social Science Planning Fund [L21CGL021].  

Reviewers' comments:

Reviewer's Responses to Questions

**Comments to the Author**

1. Is the manuscript technically sound, and do the data support the conclusions?

Reviewer #1: Yes

Reviewer #2: Yes

Reviewer #3: Yes

2. Has the statistical analysis been performed appropriately and rigorously? 

Reviewer #1: Yes

Reviewer #2: Yes

Reviewer #3: Yes

3. Have the authors made all data underlying the findings in their manuscript fully available?

Reviewer #1: Yes

Reviewer #2: Yes

Reviewer #3: Yes

4. Is the manuscript presented in an intelligible fashion and written in standard English?

Reviewer #1: Yes

Reviewer #2: Yes

Reviewer #3: Yes

5. Review Comments to the Author

Reviewer #1: The paper develops an approach of evaluating and optimizating the service resource configuration in the context of personalized customization, where an evaluation method for the relative importance of customer requirements and improved genetic algorithm for the service resource configuration were proposed, and the viability and efficacy of the approach were demonstrated by an example of living room customization in a customization company. The research has important significance in both theory and practice for the for the importance analysis of customer requirements, but it still has some issues that the authors need to address to improve the quality of the paper.

(1) In the main text, there is some error for the format of the reference.

(2) Some writing error exists in the paper. Please check the full text carefully and revise them.

In Eq. 7, what is ‘wA-l’? Should it be ‘wZ-l’.

In page 28, ‘As shown in Fig. 7, this paper sets the parameters of the original GA involved in the comparison to two cases,’ the ‘Fig.7’ may be ‘Fig. 11’.

(3) In the case study, the example studied the living room with an area of 30 square meters and located in Shenyang City, Liaoning Province, China. Besides, The author stated that they investigated 148 adult customers to obtain related data used in this paper. Please give a detailed illustration about these 148 customers to testify the rationality of selecting these 148 adult customers, such as the relationship between them.

(4) In table 12, how did the authors determine the value of the parameters illustrated in section 4.2? For example, the ‘g1’ in Eq. 8 is valued as ‘g1=2’ in table 12, and so on.

(5) In section 6.5, the author provided the performance analysis by comparison result of IGA and GA, where original GA with two different parameter settings. However, these two parameter settings are different with the one used in IGA. The author is advised to add an experimental comparison with GA that has the same parameter settings used in IGA shown in Table 14 to increase the credibility of the results.

Reviewer #2: The paper addresses an important topic in service resource configuration optimization, focusing on incorporating customer requirements and a "one-to-many" relationship between service modules and candidate itemsets. Additionally, the proposed approach integrating multiple methods (importance judgment matrix, KANO model, House of Quality, etc.) appears novel and well-reasoned.

Areas for Improvement:

- Provide a more brief real-world example early on to illustrate the practical significance of this research problem.

- The description of the proposed approach in Section 4 could benefit from a high-level overview or flowchart before diving into the detailed steps.

- How might this method be adapted or extended to account for dynamic customer preferences that may change over time?

- Elaborate on the broader implications and potential applications of this research in other domains beyond living room customization.

- In what ways could machine learning or artificial intelligence techniques be integrated into this approach to further enhance its capabilities?

- Break up some of the longer paragraphs into shorter ones to improve readability.

Reviewer #3: This paper presents a customer-centric method for optimizing service resource configuration in personalized customization. By integrating an importance judgment matrix, the KANO model, and a competitiveness evaluation, it evaluates customer requirements and their influence on service modules. Using the House of Quality (HoQ) model, the study maps customer needs to technical attributes to identify optimal service configurations. The feasibility of this approach is demonstrated through a living room customization case study, utilizing an improved genetic algorithm (IGA). The results indicate enhanced customer satisfaction through more tailored resource allocation strategies.

Point-by-Point Review:

1.The title effectively reflects the content and focus of the research. The abstract is informative, providing a concise summary of the problem, proposed solution, and methodologies used. It mentions key elements such as the KANO model, HoQ, and genetic algorithm, all relevant to the topic. However, the readability of the abstract could be improved for better flow.

2.The introduction provides a strong foundation by addressing the growing importance of personalized customization in today's market. It clearly explains the shift from a product-oriented to a service-oriented business model. However, some parts are overly verbose, and there is some repetition in discussing the problem and research objectives. The identification of the research gap—previous studies lacking a "one-to-many" relationship mechanism between service modules and candidate itemsets—is particularly strong. The objectives are clear, though the introduction would benefit from a more explicit research question or hypothesis.

3.The review covers the relevant literature on service requirement analysis, modularization, and service configuration. However, some references appear outdated or tangential to the core contribution. The research gap is well-articulated, though the review could better highlight how the proposed work advances or differs from existing approaches.

4.The paper introduces the importance judgment matrix, the KANO model, and the HoQ, which are well-suited for evaluating customer needs and service modules. However, the explanation of the mathematical model could be more concise and accessible for readers unfamiliar with these techniques. The use of an improved genetic algorithm (IGA) is well-justified, particularly for handling large-scale optimization. Further details on how the IGA improves upon traditional genetic algorithms would add clarity. The case study on living room customization is useful, though breaking it into smaller, more manageable sections would improve reader comprehension. I would recommend including a more straightforward or introductory example before delving into the living room customization example. This would help readers grasp the core concepts before dealing with a more complex scenario.

5.The quantitative analysis appears thorough, though some aspects of the results section seem rushed. Greater emphasis on the real-world implications of the results, particularly customer satisfaction, would strengthen the analysis. The inclusion of various utility functions is commendable, but clearer justification for the selection of specific functions for different service modules would aid in understanding.

6.The discussion provides insightful reflections on the model's effectiveness. However, it could benefit from more direct comparisons to alternative approaches from the literature. The paper does not explicitly address the limitations of the approach. A discussion of potential weaknesses or areas for improvement would enhance the rigor of the analysis—such as scalability to other industries including a brief discussion of computational costs or the complexity of scaling up the model. It could also benefit from a discussion of specific user-centric metrics or KPIs (Key Performance Indicators): how are the improvements in customer satisfaction quantitatively measured and compared to other methods? You may also consider adding a discussion on how the model could adapt to emerging trends in customization, such as the use of AI in real-time customization, or integration with cloud-based services for better data management. Future-proofing the methodology could increase its long-term relevance.

7.The conclusion effectively summarizes the key contributions of the paper but could be more succinct. Suggestions for future research are implied but not fully developed. Expanding on specific areas for further exploration, such as integrating real-time data or applying the model to different customization domains, would strengthen the conclusion.

---

Summarization of the main recommendations to consider when applicable:

- Clarity: Simplify some sections, especially the methodology and case study, to ensure they are accessible to a broader audience. Technical terms like "candidate itemsets" or "importance judgment matrix" may not be familiar to all readers so that clarifying them upfront would improve accessibility.

- Consistency: Ensure consistent use of key terms (e.g., service modules, candidate itemsets) throughout the paper.

- Flow: Improve the document's flow by reducing redundancy and tightening the narrative.

- Emphasize Contribution: Highlight the 'one-to-many' relationship mechanism as the primary innovation of the research.

- Figures/Tables: While the diagrams and matrices (e.g., House of Quality) are useful, adding more detailed explanations in the figure captions would enhance readability.

- Future Work: Provide more specific suggestions for future research, such as applying the model to different industries or incorporating real-time data.

- References: Ensure that all references are up-to-date and directly relevant to the research topic.

6. PLOS authors have the option to publish the peer review history of their article (what does this mean? ). If published, this will include your full peer review and any attached files.

**Do you want your identity to be public for this peer review?** For information about this choice, including consent withdrawal, please see our Privacy Policy .

Reviewer #1: No

Reviewer #2: **Yes: ** Surendra Pandey

Reviewer #3: No

---

## [Author Response · Author response to Decision Letter 1]

9 Dec 2024

Dear reviewers, we thank you very much for the comments and suggestions. They are very helpful for us to revise and improve the paper. The paper has been carefully revised according to the referee’s advice. Thank you.

---

## [Decision Letter · Decision Letter 1]

18 Feb 2025

Personalized customization: service resource configuration optimization driven by customer requirements accurately

PONE-D-24-37417R1

Dear Dr. Wang,

We’re pleased to inform you that your manuscript has been judged scientifically suitable for publication and will be formally accepted for publication once it meets all outstanding technical requirements.

Kind regards,

Sivakumar Poruran

Academic Editor

PLOS ONE

Additional Editor Comments (optional):

Reviewers' comments:

Reviewer's Responses to Questions

**Comments to the Author**

1. If the authors have adequately addressed your comments raised in a previous round of review and you feel that this manuscript is now acceptable for publication, you may indicate that here to bypass the “Comments to the Author” section, enter your conflict of interest statement in the “Confidential to Editor” section, and submit your "Accept" recommendation.

Reviewer #1: All comments have been addressed

Reviewer #2: (No Response)

2. Is the manuscript technically sound, and do the data support the conclusions?

Reviewer #1: Yes

Reviewer #2: Yes

3. Has the statistical analysis been performed appropriately and rigorously? 

Reviewer #1: Yes

Reviewer #2: Yes

4. Have the authors made all data underlying the findings in their manuscript fully available?

Reviewer #1: Yes

Reviewer #2: Yes

5. Is the manuscript presented in an intelligible fashion and written in standard English?

Reviewer #1: Yes

Reviewer #2: Yes

6. Review Comments to the Author

Reviewer #1: The revised paper has been improved, and all comments have been revisoed. I think this version is suitable for publication.

Reviewer #2: 1. The authors are suggested to just make sure that the main contributions of the article are presented in a bullet point format towards the end of the introduction before the paper organization.

2. The authors are suggested to proof-read the entire paper to make sure no grammatical errors are present

3. Figure 2 should be either enlarged or separated into two to improve readability and illustration. And same with figure 3

7. PLOS authors have the option to publish the peer review history of their article (what does this mean? ). If published, this will include your full peer review and any attached files.

**Do you want your identity to be public for this peer review?** For information about this choice, including consent withdrawal, please see our Privacy Policy .

Reviewer #1: No

Reviewer #2: No

---

## [Editor Report · Acceptance letter]

PONE-D-24-37417R1

PLOS ONE

Dear Dr. Wang,

I'm pleased to inform you that your manuscript has been deemed suitable for publication in PLOS ONE. Congratulations! Your manuscript is now being handed over to our production team.

Kind regards,

on behalf of

Dr. Sivakumar Poruran

Academic Editor

PLOS ONE